# Dominant immune tolerance in the intestinal tract imposed by RelB-dependent migratory dendritic cells regulates protective type 2 immunity

Anna-Lena Geiselhöringer [1], Daphne Kolland[1], Arisha Johanna Patt [1], Linda Hammann[2], Amelie Köhler[1], Luisa Kreft [1,12], Nina Wichmann [1], Miriam Hils[3], Christiane Ruedl [4], Marc Riemann[5], Tilo Biedermann [3], David Anz[2,6], Andreas Diefenbach [7,8], David Voehringer [9], Carsten B. Schmidt-Weber [1,10], Tobias Straub [11], Maria Pasztoi [1] & Caspar Ohnmacht [1] ✉

Dendritic cells (DCs) are crucial for initiating protective immune responses and have also been implicated in the generation and regulation of Foxp3+ regulatory T cells (Treg cells). Here, we show that in the lamina propria of the small intestine, the alternative NF-κB family member RelB is necessary for the differentiation of cryptopatch and isolated lymphoid follicle-associated DCs (CIA-DCs). Moreover, single-cell RNA sequencing reveals a RelB-dependent signature in migratory DCs in mesenteric lymph nodes favoring DC-Treg cell interaction including elevated expression and release of the chemokine CCL22 from RelB-deficient conventional DCs (cDCs). In line with the key role of CCL22 to facilitate DC-Treg cell interaction, RelB-deficient DCs have a selective advantage to interact with Treg cells in an antigen-specific manner. In addition, DC-specific RelB knockout animals show increased total Foxp3+ Treg cell numbers irrespective of inflammatory status. Consequently, DC-specific RelB knockout animals fail to mount protective Th2-dominated immune responses in the intestine after infection with *Heligmosomoides polygyrus bakeri*. Thus, RelB expression in cDCs acts as a rheostat to establish a tolerogenic set point that is maintained even during strong type 2 immune conditions and thereby is a key regulator of intestinal homeostasis.

Type 2 immunity plays a pivotal role in fighting parasitic infections but contributes also to the pathogenesis of allergic disorders. Not surprisingly, both scenarios manifest particularly at barrier sites such as the skin, the lung, or along the intestinal tract as exposure to allergens and parasites preferentially occurs at these sites. In the intestinal tract, much has been learned in recent years about early events leading to the initiation of innate type 2 immunity ranging from the discovery of type 2 innate lymphoid cells (ILC2s) and the activation of eosinophils to the activation of tuft cells[1]. However, it is still only partially understood how these early events are translated into robust T cell immunity including the formation of type 2 T helper (Th2) effector cells and T cell memory formation. Noteworthy, the intestinal tract is also considered highly tolerogenic due to the necessity of T helper (Th) cells to tolerate diverse dietary and microbial antigens. In fact, the absence of microbial

or dietary antigens predisposes the intestinal tract to Th1- or Th2-biased immunity, respectively[2], and intestinal bacteria-derived products can foster regulatory T cell (Treg) induction[3]. Since Treg cells in the gastrointestinal tract are known to control type 2 immunity[4,5], Treg cell numbers represent a critical checkpoint for establishing tolerance and restricting immune responses.

Dendritic cells (DCs) represent the most potent antigen-presenting cell type (APC) that can induce tolerogenic but also diverse effector Th cell responses. In the absence of DCs, neither tolerance towards dietary antigens[6] nor a proper Th2 response against helminth parasites or allergens can be mounted[7–9]. DCs can be sub-divided into plasmacytoid DCs thought to act primarily via cytokine secretion, e.g. upon virus infection, whereas conventional DCs (cDCs) mount robust Th1, Th2 and Th17 effector cell responses via a combination of T cell receptor stimulation, co-stimulation and secreted cytokines. Under certain circumstances such as in tumors or in the intestinal tract upon exposure to harmless antigens, cDCs may also induce differentiation of Foxp3+ Treg cells[9]. Previous studies investigated whether specialized cDC subsets are responsible for different types of Th cell responses[10]. Based on the expression of cell-surface receptors and transcription factors, cDCs have been divided into two major subsets called cDC1 (e.g. Xcr1, Irf8, Batf3) and cDC2 (e.g. Sirpα, CD11b, Irf4). cDC1 drive Th1 cell differentiation and CD8+ T cell activation via cross-presentation of soluble antigens whilst cDC2 drive Th2 and Th17 cell immune responses but may also, together with cDC1s, drive Treg cell maintenance and differentiation[11,12]. The Treg cell-inducing capacity of cDCs may be fine-tuned by environmental signals because the Treg cell-inducing capacity of Flt3-induced DCs depends on age and can be further manipulated by microbial metabolites like butyrate via change of gene expression patterns, for example by downregulation of the NF-κB family member *RelB*[3,13].

As cDC1 and cDC2 may fulfill different immunological functions, cDC subset ratios may influence Th cell responses and vary between different organs. cDC2 typically outnumber cDC1 in most murine and human tissues and comprise a considerably higher heterogeneity compared to cDC1[14,15]. Whether individual cDC2 subsets have unique functions in terms of T cell priming is still a matter of debate as genetic tools for specifically targeting cDC2 are not available to date[16]. Still, the requirement of unique transcription factors for the efficient priming of Th2-mediated immune responses has raised the possibility that specialized cDC2 subsets induce different immune responses[15,17]. For example, genetic deletion of Irf4, Klf4, Mgl2 and Notch2 has shown the importance of these transcription factors for the differentiation of cDC2 subsets and their capacity to mount robust Th2-driven immune responses[12,16,18]. A recent publication further suggested transcriptionally distinct DC subsets with unique functions, characterized as a mature Ccr7+ DC subset in non-small-cell lung cancer with a distinct immunoregulatory program characterized by expression of several costimulatory and maturation genes[19]. This mature DC subset expresses features of cDC1s and cDC2s and may regulate T cell functionality according to a unique tolerogenic program independent from the current cDC1 or cDC2 classification. These DCs enriched in immunoregulatory molecules (termed mregDC) were characterized by the downregulation of genes associated with Toll-like receptor signaling, while *Ccr7, Fscn1, IL4i1, Socs2, and RelB* gene expression were upregulated[19].

The non-canonical NF-κB member RelB is part of the alternative NF-κB pathway and plays a major role in the maturation of thymic epithelial cells (mTECs). Full RelB-deficiency therefore results in systemic T cell-dependent autoimmunity[20]. In contrast to most other autoimmune syndromes associated with a lack of central tolerance, RelB deficiency shifts autoimmune Th cells towards Th2 cell differentiation, and such mice suffer from elevated IgE levels, atopic dermatitis-like syndrome and exaggerated lung allergy[21,22]. This raises the question of whether RelB regulates Th2 cell differentiation in a cell-intrinsic or -extrinsic manner beyond incomplete negative selection in the thymus. Noteworthy, lung allergy can be resolved by the transfer of RelB-competent DCs suggesting that the expression of RelB in DCs is necessary to prevent lung inflammation in this setting[21]. During DC development, RelB may contribute to the terminal differentiation of cDC2s from a committed cDC2 precursor, as RelB overexpression in human hematopoietic progenitors led to an increase in monocytic CD14+CD11b+ precursor cells of interstitial DCs[23]. In adult animals, only a subset of splenic cDC2s is reduced when RelB is genetically deleted in DCs but whether DCs in non-lymphoid tissues are also affected remains unknown[24]. In the skin, Ccr7+CD40high migratory DCs have been shown to express RelB[25] and upon RelB ablation, the proliferation of (self-reactive) Foxp3+ Treg cells in the skin was increased[26]. We and others have recently described that RelB-deficiency in DCs is also associated with a slight type 2 immune bias with elevated frequencies of GATA3+ Th2 cells in the intestinal tract and elevated IgE serum levels at steady-state. Furthermore, these mice showed a 50% reduction of microbiome-induced RORγt-expressing Treg cells in the intestinal tract while at the same time showing a systemic, vigorous expansion of Foxp3+ Treg cells ultimately protecting these animals from autoimmune diseases[27]. This observation led us to the question whether RelB affects DC numbers or transcriptional profiles of DC subsets in the small intestine and gut-draining mesenteric lymph nodes (mLN) and whether bystander immune tolerance by such an expanded Treg cell population can control type 2 immunity in allergic and parasitic immune responses.

Here, we provide direct evidence that RelB deficiency in CD11c+ cells neither affects DC numbers nor their major subset distribution in the lamina propria of the small intestine (SI-LP) and mLN. However, single-cell RNA-sequencing (scRNAseq) revealed that a DC subset surrounding cryptopatches and isolated lymphoid follicles (CIA-DCs) was completely absent in SI-LP of *Itgax*Cre × *RelB*fl/fl (RelBΔDC) mice. Yet, the sole CIA-DC deficiency did not affect Treg cell frequencies nor Treg subset cell distribution. In contrast to the limited effect of RelB in the SI-LP, migratory but not resident DCs from mLN showed a distinct RelB-dependent clustering including different expression of co-stimulatory molecules, chemokines and genes associated with migration and other tolerogenic markers. RelB deficient DCs were more likely to interact with Treg cells compared to RelB-competent DCs. While RelBΔDC mice were not more susceptible to food allergy infection with the helminth parasite *Heligmosomoides polygyrus bakeri* (*Hpb*) resulted in a restrained Th2 cell response in RelBΔDC mice that led to an impaired Treg cell-dependent protection from a secondary infection. Thus, RelB-deficiency in DCs promote a dominant immune tolerance via maintaining a robust increase in Treg cells even under strong Th2 cell-polarizing conditions. Our work highlights the power of RelB as a molecular switch for a regulatory transcriptional profile in migratory cDC2s to maintain robust immune tolerance even under strong inflammatory conditions.

## Results

### RelB deficiency in dendritic cells affects intestinal Treg and Th2 cells but only marginally resident and migratory DC numbers

To investigate the consequences of RelB deletion in CD11c+ cells at mucosal surfaces, we first characterized Th cells and DCs of RelBΔDC mice in mLN and the SI-LP by flow cytometry. As shown before for other organs[27], RelBΔDC animals showed an increase in total Foxp3+ Treg cells in mLN and the SI-LP (Fig. 1a, b and Supplementary Fig. 1a, b). The increase in Foxp3+ Treg cells was primarily due to Treg cells co-expressing GATA3 while RORγt+ Treg cells were decreased by approx. 50% in both organs (Fig. 1c, d and Supplementary Fig. 1c, d)). For T effector cells, no difference was found in lamina propria RORγt+ Th17 cells (Fig. 1f and Supplementary Fig. 1f). However, we found a moderate but very consistent increase in the frequency of GATA3+ Th cells in both the mLN and the SI-LP (Fig. 1e, f and Supplementary Fig. 1e, f)

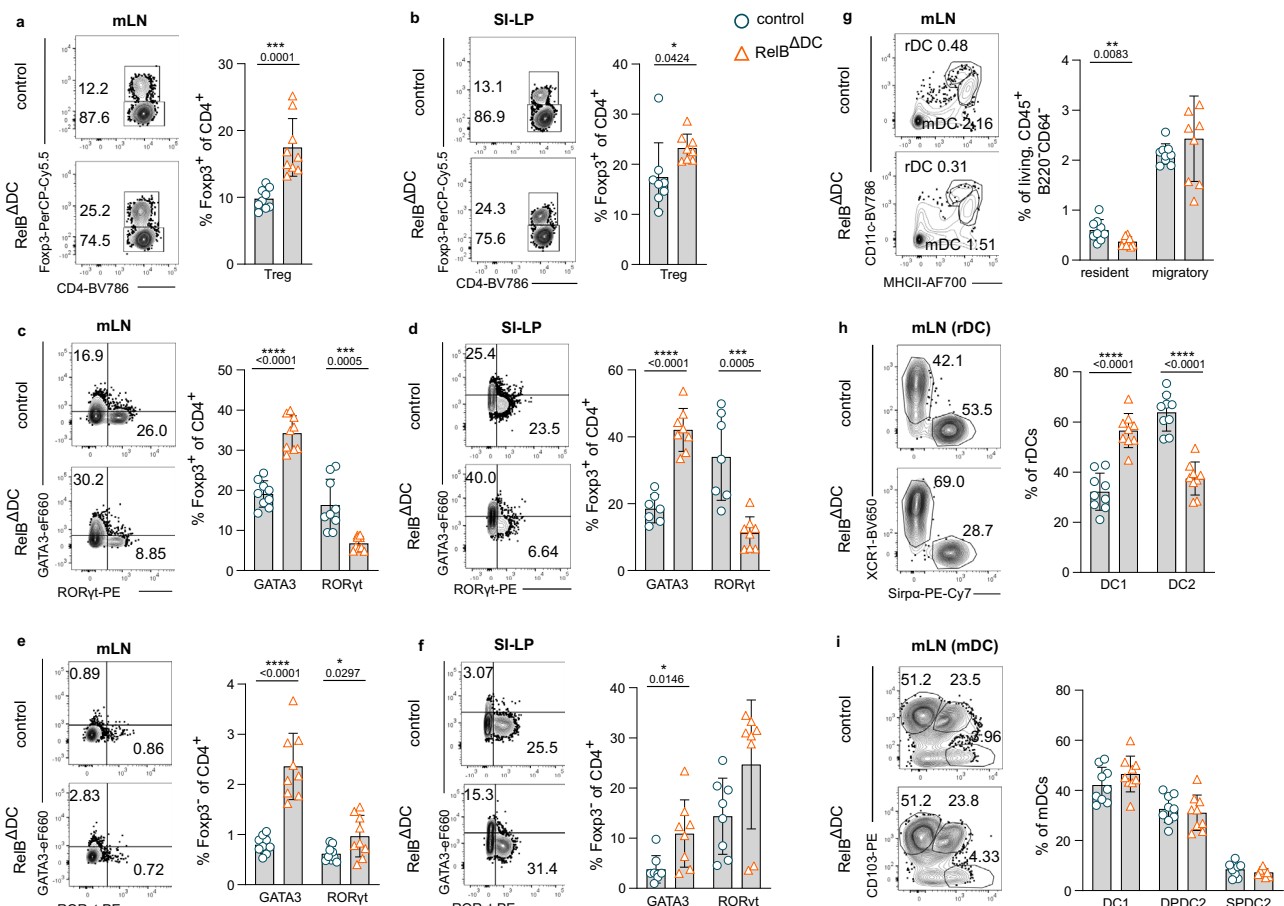

**Fig. 1 | RelB deficiency in CD11c⁺ cells results in a reduction of resident cDC2s in gut-draining mesenteric lymph nodes and elevated GATA3⁺ T and Treg cells.** **a–f** Flow cytometric analysis of T cell populations in mesenteric lymph nodes (mLN) and the lamina propria of the small intestine (SI-LP) of control and RelBΔDC mice at steady-state. **a, b** Representative flow cytometry plots (left) and quantification (right) of frequency of Foxp3⁺ Treg cells among CD4⁺ T cells in mLN (**a** control $n = 9$, RelBΔDC $n = 9$) and SI-LP (**b** control $n = 8$, RelBΔDC $n = 8$). **c, d** Representative flow cytometry plots (left) and quantification (right) of frequency of GATA3- and RORγt-expressing Foxp3⁺ Treg cells in mLN (**c** control $n = 9$, RelBΔDC $n = 9$) and SI-LP (**d** control $n = 7$, RelBΔDC $n = 8$). **e, f** Representative contour plots (left) and quantification (right) of frequency of GATA3 and RORγt expression in Foxp3⁻ CD4⁺ Th cells in mLN (**e**, control $n = 9$, RelBΔDC $n = 9$) and SI-LP (**f** control $n = 8$, RelBΔDC $n = 8$). **g–i** Flow cytometric analysis of dendritic cell populations in mLN of control and RelBΔDC mice

at steady-state. **g** Representative contour plots (left) of resident cDCs (rDCs; live dead⁻CD45⁺B220⁻CD64⁻CD11cʰⁱᵍʰMHCIIⁱⁿᵗ) and migratory cDCs (mDCs; live dead⁻CD45⁺B220⁻CD64⁻CD11cⁱⁿᵗMHCIIʰⁱᵍʰ) (left) and quantification of indicated frequencies (right) in mLN at steady-state (control $n = 9$, RelBΔDC $n = 9$). **h** Representative contour plots (left) of resident DCs subsets (rDC1: XCR1⁺Sirpα⁻; rDC2: XCR1⁺Sirpα⁺) and frequencies (right) in mLN (control $n = 9$, RelBΔDC $n = 9$). **i** Representative contour plots (left) of migratory DC subsets (mDC1: CD103⁺CD11b⁻ double-positive DCs (DPDC2): CD103⁺CD11b⁺; single-positive DCs (SPDC2): CD103⁻CD11b⁺) and frequencies (right) in mLN (control $n = 9$, RelBΔDC $n = 9$). Each dot represents an individual mouse and mean ± SD from two independent experiments is shown. Statistical analysis was performed using two-tailed students $t$-test. $P$ value of <0.05 was considered statistically significant with *$p < 0.05$, **$p < 0.01$, ***$p < 0.001$, ****$p < 0.0001$. Source data are provided as a Source Data file.

suggesting a common underlying mechanism for the regulation of GATA3-expressing Treg and Th cells in the absence of RelB in CD11c⁺ cells. Noteworthy, we did not observe any difference in Th cells producing type 2 cytokines IL-4 or IL-13 nor did Treg cells produce relevant amounts of these cytokines (Supplementary Fig. 1g, h). However, we observed slightly reduced frequencies and numbers of IL-10-secreting Treg cells in mLN of RelBΔDC mice possibly due to a higher activation state (Supplementary Fig. 1h). Since GATA3 expression was proposed to be a hallmark of Treg cells isolated from non-lymphoid tissue[28], we also measured the frequencies of precursors for non-lymphoid tissue Treg cells residing in secondary lymphoid organs[29]. Indeed, we found elevated frequencies of stage 2 and stage 3 tissue Treg precursor cells in mLN of RelBΔDC mice compared to littermate control animals (Supplementary Fig. 1i–k). This observation indicates that DCs might constantly transport self-antigens from tissues to draining lymph nodes to induce differentiation and be responsible for the observed effects on Treg cells. Therefore, we raised the question

whether RelB deficiency affects DC cell numbers or their overall subset distribution in either gut-draining mLN or directly in the lamina propria.

Notably, RelB ablation in CD11c⁺ cells did not affect the overall frequency or cell number of DCs in mLN or the SI-LP (Supplementary Fig. 1l, m) nor was DC subset distribution affected in the lamina propria (Supplementary Fig. 1m). Similarly, mLN of RelBΔDC mice harbored equal numbers of migratory and resident cDCs even though the latter tended to be slightly lower in frequency and number which was primarily due to a reduced frequency and cell number of resident cDC2s and is consistent with previous findings (Fig. 1g, h and [30]). Importantly, the frequency and number of migratory DCs (CD11cⁱⁿᵗMHC-IIʰⁱ) or their subsets were not affected by RelB ablation (Fig. 1i). Thus, we concluded that RelB ablation in CD11c⁺ cells alters the steady-state Th cell distribution but had only a slight effect on resident cDC2s in mLN while not affecting the cell numbers of other DC subsets in the mLN or the lamina propria.

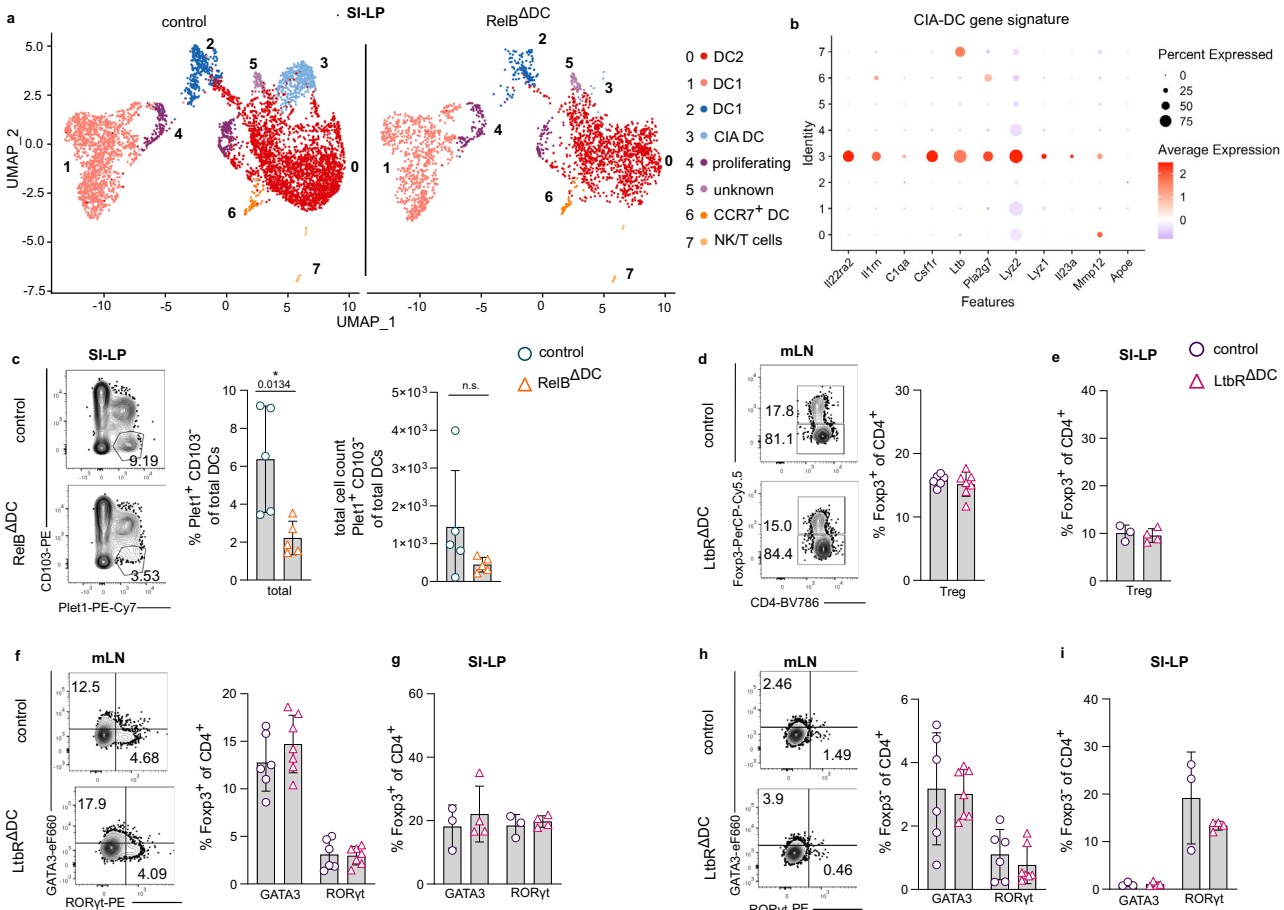

**Fig. 2 | RelB is crucial for the differentiation of CIA-DCs in the lamina propria.**
**a** UMAP projection of scRNAseq gene expression profiling from CD11c⁺MHCII^high DCs (among live/dead⁻CD45⁺B220⁻CD64⁻) sorted from the lamina propria of the small intestine (SI-LP) of steady-state control (left) and RelB^ΔDC (right) mice.
**b** Comparison of CIA-DC signature genes across clusters identified in (**a**).
**c** Representative flow cytometry plots of Plet1⁺CD103⁻ DCs (cryptopatch and isolated lymphoid follicle-associated DCs; CIA-DCs) in SI-LP of control and RelB^ΔDC mice at steady-state (left), quantification of frequencies (middle) and total cell numbers (right) (control $n = 5$, RelB^ΔDC $n = 5$). **d–i** Flow cytometric analysis of T cell populations in mesenteric lymph node (mLN) and SI-LP of control and LtbR^ΔDC mice at steady-state. **d, e** Representative flow cytometry plots (left) and quantification (right) of Foxp3⁺ Treg cell frequencies among CD4⁺ T cells in mLN (**d**, control $n = 6$,

LtbR^ΔDC $n = 7$) and SI-LP (**e** control $n = 3$, LtbR^ΔDC $n = 4$). **f, g** Representative flow cytometry plots (left) and quantification of frequencies (right) of GATA3- and RORγt-expressing Foxp3⁺ Treg cells in mLN (**f** control $n = 6$, LtbR^ΔDC $n = 7$) and SI-LP (**g** control $n = 3$, LtbR^ΔDC $n = 4$). **h, i** Representative contour plots (left) and quantification of frequencies (right) for GATA3 and RORγt expression among Foxp3⁻CD4⁺ Th cells in mLN (**h** control $n = 6$, LtbR^ΔDC $n = 7$) and SI-LP (**i** control $n = 3$, LtbR^ΔDC $n = 4$). scRNAseq experiments were performed from sort-purified DCs pooled from three individual mice per genotype. Each dot represents an individual mouse and mean ± SD of two independent experiments is shown. Statistical analysis was performed using two-tailed students $t$-test. $P$ value of <0.05 was considered statistically significant with *$p < 0.05$, **$p < 0.01$, ***$p < 0.001$, ****$p < 0.0001$, n.s. not significant. Source data are provided as a Source Data file.

## RelB-dependent cryptopatch and follicle-associated DCs (CIA-DCs) in the lamina propria are not responsible for the Treg cell increase in RelB^ΔDC mice

So far, our results showed a shift in the cDC1 to cDC2 ratio in resident DCs of mLN, but no major effects of RelB depletion in DCs of migratory subsets. To verify these findings and make use of a less biased approach, we decided to further characterize RelB-dependent transcriptional profiles by scRNAseq of all CD11c^hiMHC-II⁺ cells isolated from mLN and SI-LP. CD11c^hiMHC-II⁺ cells from the lamina propria clustered into 7 distinct cell populations with clusters 1−6 expressing DC genes and cluster 7 containing contaminating NK/T cells. Not surprisingly, the major clusters were identified as cDC1 (cluster 1) and cDC2s (cluster 0) that were identical in distribution and gene expression profiles when comparing DCs from RelB-sufficient and RelB-deficient animals (Fig. 2a and Supplementary Fig. 2a−e). We only noticed one exception: A small population of cells (cluster 3) was completely absent in RelB^ΔDC mice (Fig. 2a). This cluster showed a gene

expression profile that was reminiscent of a recently described subset of DCs (CIA-DCs) localizing around cryptopatches (CPs) and isolated lymphoid follicles (ILFs) including high expression of *Il22ra2, Csf1r, IL23, Pla2g7* and *Lyz2* (Fig. 2b and [31]). Flow cytometry-based identification of CIA-DCs with an anti-Plet1 antibody[31,32] confirmed a strong reduction of Plet1⁺CD103⁻ CIA-DCs in SI-LP of RelB^ΔDC mice (Fig. 2c). CIA-DCs have been shown to depend on lymphotoxin (LT) β receptor signaling and innate lymphoid cell type 3- (ILC3s) and/or lymphocyte-derived LTβ may contribute to CIA-DC differentiation/maintenance[31]. Interestingly, we found robust LTβ expression in DCs of cluster 3 (Fig. 2b). Since LTβ signaling is known to induce alternative NF-κB signaling including RelB[33,34], also DC-derived LTβ may contribute to CIA-DC differentiation in an autocrine and RelB-dependent manner. To investigate whether the absence of CIA-DCs contributes to the accumulation of intestinal Tregs, we made use of mice with a specific deletion of LTβR in CD11c⁺ cells (LtbR^ΔDC mice) that selectively lack CIA-DCs[31]. There was no difference in the overall frequency or absolute cell

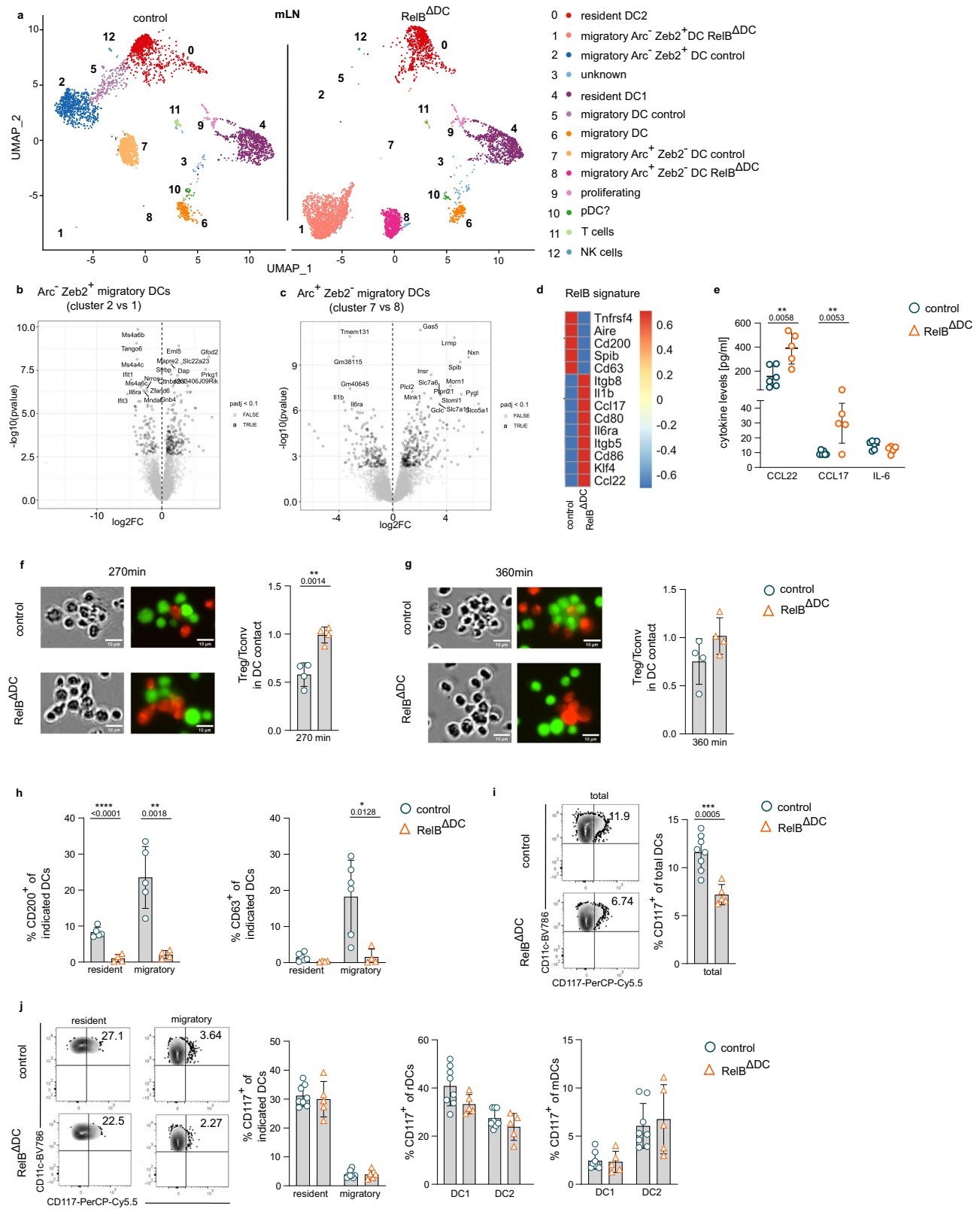

number of Foxp3[+] Treg cells in mLN or SI-LP nor did we observe a difference in the frequency or absolute cell number of GATA3[+]/RORγt[+] Treg cell subsets or GATA3[+]/RORγt[+] T effector cells in LtbR[ΔDC] mice (Fig. 2d–h and Supplementary Fig. 2f–k). In summary, these results suggest that RelB regulates the differentiation of CIA-DCs but the absence of CIA-DCs does not explain the accumulation of intestinal Treg cells in RelB[ΔDC] mice.

## RelB controls transcriptional profiles of migratory DCs in mesenteric lymph nodes

In contrast to DCs from the lamina propria, DCs from mLN clustered into 11 DC clusters (and two contaminating ILC/T cell clusters; clusters 11 and 12, respectively) (Fig. 3a). Again, a few clusters representing resident cDC1 (cluster 4) and resident cDC2 (cluster 0) based on the absence of the lymph node homing receptor *Ccr7* were equally

**Fig. 3 | RelB regulates gene expression of migratory DCs in the mesenteric lymph node. a** UMAP projection of scRNAseq gene expression profiling from CD11c$^+$MHCII$^{hi}$ DCs (among live/dead$^-$CD45$^+$B220$^-$CD64$^-$) sorted from mesenteric lymph node (mLN) of naïve control (left) and RelB$^{ΔDC}$ (right) mice. **b** Volcano plot of differentially expressed genes comparing Arc$^-$Zeb2$^+$ migratory DCs from cluster 2 (control) and cluster 1 (RelB$^{ΔDC}$). **c** Volcano plot of differentially expressed genes comparing Arc$^+$Zeb2$^-$ migratory DCs from cluster 7 (control) and cluster 8 (RelB$^{ΔDC}$). **d** Heatmap depicting a RelB-dependent core signature in CCR7$^+$ DCs in mLN. **e** Levels of Ccl22, Ccl17, and IL-6 in supernatant of CD11c-enriched cells cultured for 16 h in the absence of any stimulation (control n = 6, RelB$^{ΔDC}$ n = 5). **f, g** Representative cluster formation (phase contrast) and fluorescence images (Treg cells: red; non-Treg cells: green) (left) and quantification (right) 270 min (**f** control n = 4, RelB$^{ΔDC}$ n = 4) and 360 min (**g** control n = 4, RelB$^{ΔDC}$ n = 4) after

coculture. **h** Expression of CD200 (left, control n = 5, RelB$^{ΔDC}$ n = 4) and CD63 (right, control n = 6, RelB$^{ΔDC}$ n = 4) in resident and migratory DCs defined in Fig. 1 from mLN of control and RelB$^{ΔDC}$ mice. **i, j** Representative contour plots (left) and quantification (right) showing the expression of CD117 in total mLN DCs (**i**) and resident and migratory DCs and their respective subsets **j** from control and RelB$^{ΔDC}$ mice at steady-state (control n = 8, RelB$^{ΔDC}$ n = 5). scRNAseq experiments were performed from sort-purified DCs pooled from three individual mice per genotype. Each dot represents an individual mouse and mean ± SD of one representative experiment (**a**–**h**) or two independent experiments (**i, j**) is shown. Statistical analysis was performed using two-tailed students t-test. P value of <0.05 was considered statistically significant with *p < 0.05, **p < 0.01, ***p < 0.001, ****p < 0.0001. Source data are provided as a Source Data file.

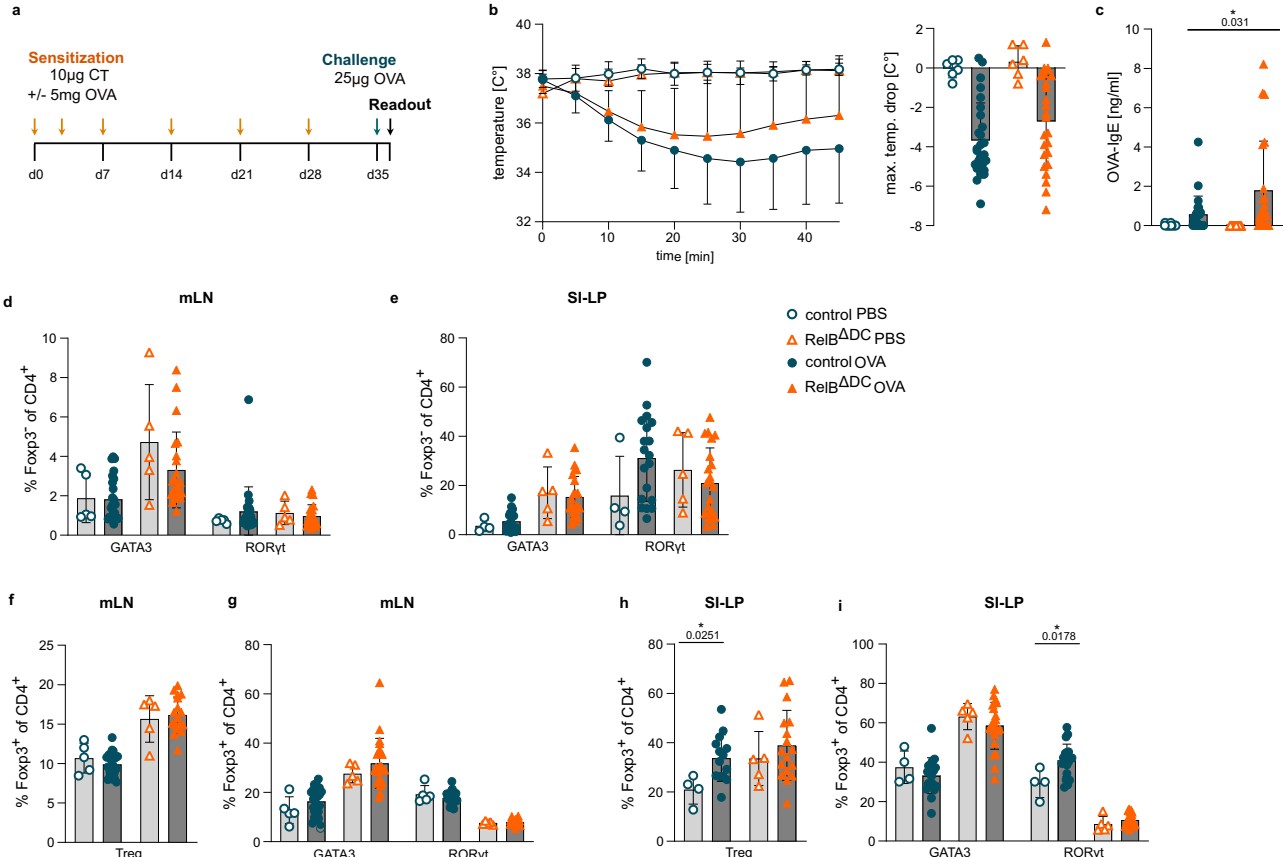

**Fig. 4 | RelB$^{ΔDC}$ mice do not show stronger anaphylaxis in a model of food allergy. a** Experimental scheme of a food allergy regime with OVA as a model antigen for sensitization. CT cholera toxin, OVA chicken ovalbumin. **b** Core body temperature drop (left) and maximal temperature drop (right) after challenge with OVA on day 35 in control and RelB$^{ΔDC}$ mice (control (PBS) n = 6, RelB$^{ΔDC}$ (PBS) n = 6, control (OVA) n = 27 and RelB$^{ΔDC}$ (OVA) n = 21/25). **c** OVA-specific IgE levels in serum in food allergic control and RelB$^{ΔDC}$ mice on day 36 (control before n = 34, RelB$^{ΔDC}$ before n = 33, control (OVA) n = 24 and RelB$^{ΔDC}$ (OVA) n = 22). **d, e** Quantification of intracellular GATA3 and RORγt expression in Foxp3$^-$ CD4$^+$ Th cells of food allergic mice in mesenteric lymph node (mLN) (**d** control (PBS) n = 5, RelB$^{ΔDC}$ (PBS) n = 5, control (OVA) n = 25 and RelB$^{ΔDC}$ (OVA) n = 22) and lamina propria of the small intestine (SI-LP) (**e** control (PBS) n = 4, RelB$^{ΔDC}$ (PBS) n = 5, control (OVA) n = 19 and RelB$^{ΔDC}$ (OVA) n = 22). **f, g** Quantification of Foxp3$^+$ CD4$^+$ T cells (**f** control (PBS)

n = 5, RelB$^{ΔDC}$ (PBS) n = 5, control (OVA) n = 20 and RelB$^{ΔDC}$ (OVA) n = 17) and intracellular GATA3 and RORγt expression in Foxp3$^+$ CD4$^+$ Tregs (**g** control (PBS) n = 5, RelB$^{ΔDC}$ (PBS) n = 5, control (OVA) n = 25/20 and RelB$^{ΔDC}$ (OVA) n = 22/17) in mLN of food allergic animals. **h, i** Quantification of Foxp3$^+$ CD4$^+$ T cells (**h** control (PBS) n = 4, RelB$^{ΔDC}$ (PBS) n = 5, control (OVA) n = 14 and RelB$^{ΔDC}$ (OVA) n = 17) and intracellular GATA3 and RORγt expression in Foxp3$^+$ CD4$^+$ Tregs (**i** control (PBS) n = 4, RelB$^{ΔDC}$ (PBS) n = 5, control (OVA) n = 19 and RelB$^{ΔDC}$ (OVA) n = 22) in SI-LP of food allergic animals. Each dot represents an individual mouse and mean ± SD from two to four independent experiments is shown. Statistical analysis was performed using two-tailed students t-test. P value of <0.05 was considered statistically significant with *p < 0.05, **p < 0.01, ***p < 0.001, ****p < 0.0001. Source data are provided as a Source Data file.

distributed among DCs from both RelB$^{ΔDC}$ and control animals but certain DC clusters exhibited a genotype-specific clustering according to the presence or absence of RelB (Fig. 3a). In line with the flow cytometric enumeration of DC subsets in both mouse lines (Fig. 1g–l and Supplementary Fig. 1l, m) the relative cluster distribution did not differ due to genotype (see also Fig. 6b). In contrast to flow cytometry-

based definition of DC subsets, clusters uniquely derived from control or RelB$^{ΔDC}$ origin revealed a migratory phenotype based on *Ccr7* expression (Supplementary Fig. 3a). Noteworthy, *Ccr7$^+$* clusters 5, 7, and 2 were exclusively found in control animals while *Ccr7$^+$* clusters 8 and 1 were only present among RelB-deficient DCs suggesting a RelB-dependent gene signature sufficiently distinct to allow individual

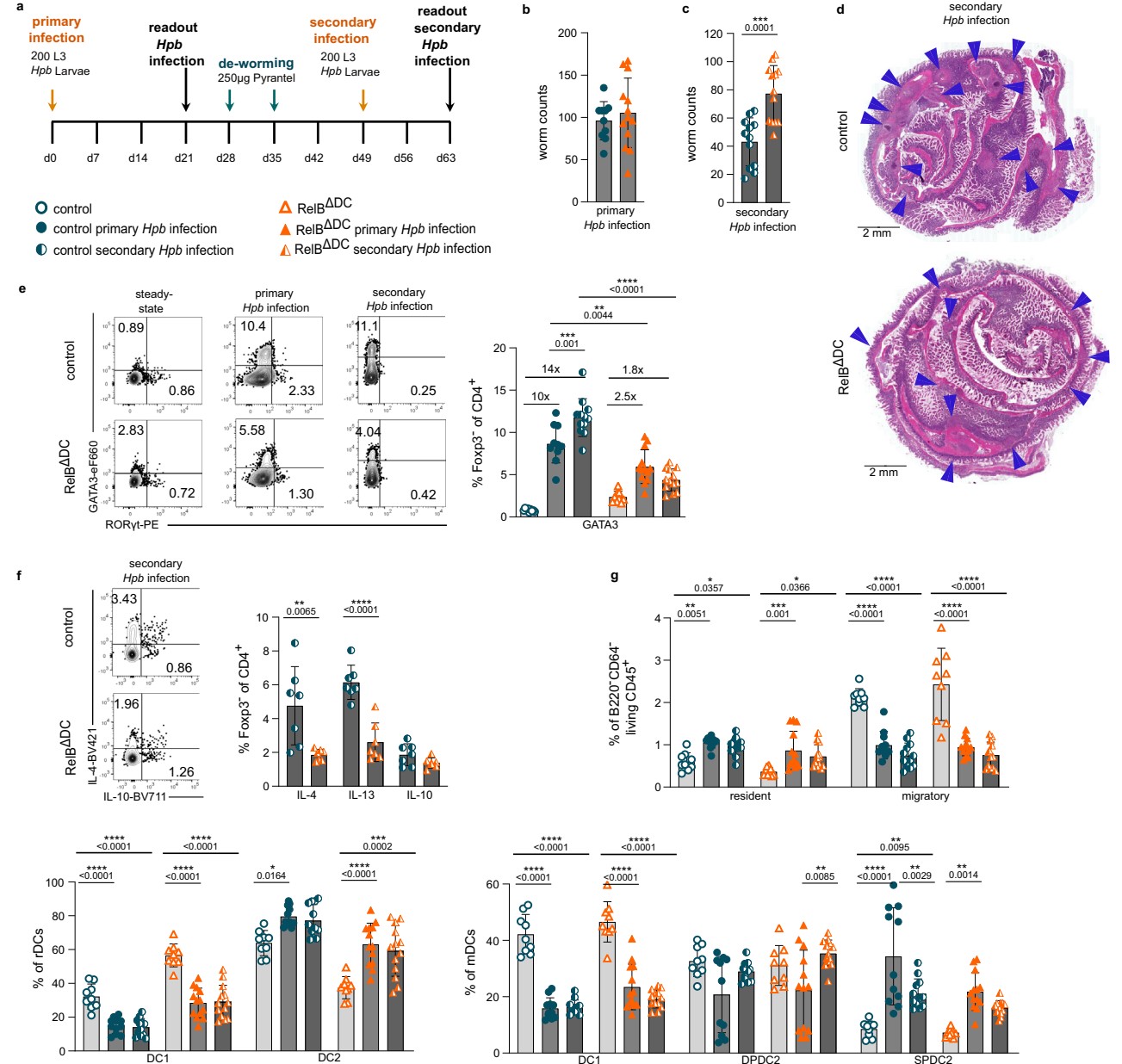

**Fig. 5 | RelB^ΔDC mice show impaired Th2 effector cell differentiation and parasite clearance after *Hpb* infection. a** Experimental scheme of primary *Heligmosomoides polygyrus bakeri* (*Hpb*) infection and *Hpb* re-infection (secondary infection) regimen. **b**, **c** Intestinal worm counts after primary (day 21, panel **b**, control (primary *Hpb* infection) n = 10 and RelB^ΔDC (primary *Hpb* infection) n = 13) and secondary (day 63, panel **c**, control (secondary *Hpb* infection) n = 12 and RelB^ΔDC (secondary *Hpb* infection) n = 13) *Hpb* infection of control and RelB^ΔDC mice. **d** Representative H&E staining of small intestinal tissue sections after secondary *Hpb* infection. Arrows indicate detected granulomas. **e** Representative flow cytometry plots (left) and quantification (right) of intracellular GATA3 and RORγt expression among Foxp3^− CD4^+ Th cells from mesenteric lymph node (mLN) of control and RelB^ΔDC mice at steady-state, primary and secondary *Hpb* infection. Control (steady-state) n = 9, control (primary *Hpb* infection) n = 10, and control (secondary *Hpb* infection) n = 12. RelB^ΔDC (steady-state) n = 9, RelB^ΔDC (primary *Hpb* infection) n = 13, and RelB^ΔDC (secondary *Hpb* infection) n = 13. **f** Representative flow cytometry plots of

intracellular IL-4 and IL-10 expression (left) and quantification (right) of cytokine expression of Foxp3^− Treg cells in mLN after secondary *Hpb* infection control (secondary *Hpb* infection) n = 7 and RelB^ΔDC (secondary *Hpb* infection) n = 7).
**g** Quantification of percentage of resident (rDCs) and migratory (mDCs) cDCs (upper panel), resident DC1s and resident DC2s (lower left panel) as well as migratory DC1s, double-positive DCs (DPDC2) and single-positive DCs (SPDC2) (lower right panel) in mLN at steady-state, after primary and secondary *Hpb* infection. Control (steady-state) n = 9, control (primary *Hpb* infection) n = 11, and control (secondary *Hpb* infection) n = 12. RelB^ΔDC (steady-state) n = 9, RelB^ΔDC (primary *Hpb* infection n = 13, and RelB^ΔDC (secondary *Hpb* infection) n = 13. Each dot represents an individual mouse and mean ± SD from at least two independent experiments is shown. Statistical analysis was performed using two-tailed students *t*-test (**b**, **c**, **f**) or one-way ANOVA with Tukey correction for multiple comparison (**e**, **g**). P value of <0.05 was considered statistically significant with *p < 0.05, **p < 0.01, ***p < 0.001, ****p < 0.0001. Source data are provided as a Source Data file.

clustering (Fig. 3a). Based on cluster tree analysis and expression of key signature genes (Supplementary Fig. 3a–c), we postulated that cluster 7 (control migratory DCs) corresponds to cluster 8 (RelB^ΔDC migratory DCs) and cluster 2 (control migratory DCs) corresponds to cluster 1 (RelB^ΔDC migratory DCs). Migratory clusters 7 and 8 were further

defined by expression of *Arc* but the absence of *Zeb2* expression whereas migratory clusters 1 and 2 had an opposite expression profile for these genes (Supplementary Fig. 3a). Zeb2 has been shown to regulate the cDC1 to cDC2 ratio in the gut by affecting DC progenitors[35,36] whereas Arc is known to regulate actin dynamics and

thus migration capacities of migratory DCs[37]. Direct comparison of the respective clusters (cluster 7 vs 8 and cluster 1 vs 2) revealed different gene expression profiles associated with inflammation (*Il6ra, IL1b*) showing higher expression in the absence of RelB (Fig. 3b, c). Moreover, we found higher expression of the chemokines *Ccl22 and Ccl17* in RelB-deficient *Ccr7*[+] DCs from mLN and substantial differences in the expression of co-stimulatory molecules (*Cd200, Cd80, Cd86, Tnfrsf4*) and integrins (*Itgb5, Itgb8*) associated to a tolerogenic function and therefore defined a RelB-dependent core gene signature among *Ccr7*[+] migratory DCs (Fig. 3d). Higher expression of CCL22 and CCL17 in the absence of RelB could be validated in unstimulated CD11c-enriched DCs isolated from mLN of the respective animals (Fig. 3e). Noteworthy, the levels of the pro-inflammatory cytokine IL-6, used as pro-inflammatory signal control, remained unchanged (Fig. 3e). To determine interaction of RelB-deficient DCs with Treg cells, we loaded DCs from the mLN and spleen of RelB[ΔDC] and control animals with chicken ovalbumin (OVA), incubated them with fluorescently labeled Treg and non-Treg cells from T cell receptor (TCR) transgenic OT-II animals and then measured cluster formation over time. In line with the elevated expression of CCL22 in DCs from RelB[ΔDC] animals (Fig. 3e) and the key role of CCL22 for regulating DC-Treg interaction[38], we found a selective advantage of RelB-deficient DCs for cluster formation with Treg cells compared to RelB competent DCs at an early time point (Fig. 3f). This difference vanished with time indicating that RelB-deficient DCs were still capable to normally interact with non-Treg cells (Fig. 3g). Additionally, we validated the RelB-dependent expression of the co-stimulatory molecule CD200 by flow cytometry (Fig. 3h). The most differentially expressed gene was CD63 (Fig. 3d, g), a member of the tetraspanin superfamily. CD63 is a membrane protein of exosomes and is used as a marker for exocytosis[39]. CD63 has been further proposed to regulate DC migration[40]. In line with this notion, we found CD63 to be exclusively expressed in migratory DCs from wildtype mice (clusters 2, 6 and 7) but not migratory DCs of RelB[ΔDC] mice (clusters 1, 6 and 8) (Fig. 3a, h). Even though *Kit* (CD117) was not differentially expressed based on our scRNAseq analysis (Supplementary Fig. 3d), one report suggested that RelB regulates CD117 expression in splenic cDC2s[30]. Therefore, we decided to determine CD117 expression in mLN DCs and confirmed its reduced expression in mLN cDCs of RelB[ΔDC] mice (Fig. 3i). Interestingly, CD117 was primarily expressed by resident DCs but not migratory cDCs (Fig. 3j) in line with the reduced frequency of resident cDC2s observed in mLN of RelB[ΔDC] mice (Fig. 1g–i). Altogether, scRNAseq profiling of RelB-sufficient and RelB-deficient mLN DCs revealed major differences in the transcriptional profile of *Ccr7*-expressing migratory cDCs. Given that these differences included many genes associated with co-stimulation and a selective advantage of RelB-deficient DCs to interact with Treg cells, these data highlight the power of RelB to control regulatory gene signatures that in sum may explain the increase in overall Tregs in mLN and the intestinal tract of RelB[ΔDC] mice.

## RelB[ΔDC] mice are not more susceptible to food allergy

Our results indicate that RelB[ΔDC] mice have an intrinsic type-2 immune tone bias at barrier sites characterized by increased frequencies of GATA3[+] Th2 cells (Fig. 1e, f), elevated serum IgE antibody titers[27], and RORγt[+] Treg cells which may regulate Th2 immune responses including food allergy[4,41]. Therefore, we next asked whether this type-2 immune bias will impact the responsiveness of RelB[ΔDC] mice to food allergy, especially as GATA3[+] Treg cells may become pathogenic by releasing type 2 cytokines and contribute to food allergy in mice or humans with enhanced interleukin-4 receptor signaling[42]. For this purpose, we set up a model of food allergy towards chicken ovalbumin (OVA) via oral sensitization and intravenous challenge (Fig. 4a). Interestingly, RelB[ΔDC] mice did not show a stronger anaphylactic reaction and the maximal drop in body temperature was in tendency lower in RelB[ΔDC] mice compared to control mice (Fig. 4b). This milder response

was not due to reduced production of IgE antibodies as OVA-specific IgE antibodies were even increased in the serum of RelB[ΔDC] mice (Fig. 4c) nor due to a reversal of the Th2 cell bias of RelB[ΔDC] mice in draining lymph nodes or the SI-LP after allergen challenge (Fig. 4d, e and Supplementary Fig. 4a, b). Levels of mast cell protease-1 (MCPT-1) as a marker for mast cell degranulation were not increased in food allergic animals and not affected by genotype (Supplementary Fig. 4c). Th17 cell frequencies were not affected even though there was a tendency for increased Th17 cells upon induction of food allergy in control animals. Similarly, elevated overall Treg cell frequencies (Fig. 4f, h and Supplementary Fig. 4d, f) and the general increase of GATA3[+] Treg cells and a respective decrease in RORγt[+] Treg cells in RelB[ΔDC] mice were maintained (Fig. 4g, l and Supplementary Fig. 4e, g). Moreover, there was an increase in RORγt[+] Treg cells in control but not RelB[ΔDC] mice upon induction of food allergy (Fig. 4i). A disturbed epithelial barrier upon allergic conditions and enhanced accessibility of luminal antigens that foster both Th17 and RORγt[+] Treg cell induction under allergic conditions may explain this effect in control animals[43]. Similarly, we did not observe a major difference in the frequency of cDCs nor their subsets in the mLN or the SI-LP (Supplementary Fig. 4h–l). Thus, we concluded that RelB[ΔDC] mice do not suffer from enhanced allergic reactions upon induction of food allergy and show a rather attenuated anaphylactic response despite elevated allergen-specific IgE levels, elevated Th2 cell frequencies, and reduced RORγt[+] Treg cell frequencies at the site of sensitization. Furthermore, we hypothesized that enhanced accessibility to luminal antigens during food allergy cannot overcome the reduced RORγt[+] Treg cell counts consistently observed in RelB[ΔDC] mice even under allergic conditions.

## RelB[ΔDC] mice fail to mount a protective Th2 immune response against secondary *Heligmosomoides polygyrus bakeri* infection

Our model of food allergy only evokes a very modest Th2 immune response (Fig. 4) and food allergy may be rather regulated by specific follicular Th cell responses co-expressing type 2 cytokines to induce high-affinity IgE responses[44]. To assess whether the Th2 cell immune bias or the increased Treg cell numbers observed in steady-state RelB[ΔDC] mice would dominate in a bona-fide Th2 cell-dependent inflammation model, we infected RelB[ΔDC] and control mice with L3 larvae of the helminth parasite *Heligmosomoides polygyrus bakeri* (*Hpb*). Mice were analyzed on day 21 post infection (primary infection) or treated with an anti-helminthic drug to eradicate *Hpb* and then re-infected with *Hpb* two weeks before analysis (secondary infection) (Fig. 5a and Supplementary Fig. 5). Protective immunity towards secondary *Hpb* infection depends on Th2 (memory) cell differentiation during the primary infection[45] and constitutes therefore a suitable model to study effective induction of Th2 effector/memory cell responses. Interestingly, primary infection with *Hpb* resulted in a similar worm count in the intestines of RelB[ΔDC] and control mice (Fig. 5b) but RelB[ΔDC] mice exhibited a reduced mLN cellularity (Supplementary Fig. 5a) and a significantly higher parasite burden after secondary *Hpb* infection (Fig. 5c). In line with this observation, RelB[ΔDC] mice had lower levels of MCPT-1 in their serum when compared to control animals indicating a reduced activation of mast cells by parasite-specific IgE antibodies (Supplementary Fig. 5b). Moreover, RelB[ΔDC] mice tended to show fewer granuloma counts which may indicate a reduced capacity to trap adult *Hpb* larvae (Fig. 5d and Supplementary Fig. 5c). Foxp3[+] Treg cell frequencies remained constantly higher in mLN and SI-LP of *Hpb*-infected RelB[ΔDC] compared to control animals even under such strong inflammatory conditions (Fig. 5f and Supplementary Fig. 5d–f). Treg cells co-expressing high levels of GATA3 protein due to mutations in either the ITIM motif of the interleukin (IL) 4 receptor (IL4R)[42] or the IPEX mutation M370I[46] have been reported to become pathogenic and produce Th2 cytokines such IL-4 or IL-13 themselves. Therefore, we also investigated whether Treg cells exhibiting high GATA3 expression in RelB[ΔDC] mice acquired

the capability to generate Th2 cell cytokines after *Hpb* infection. However, we could not detect any IL-4 or IL-13 cytokine release from mLN Treg cells after secondary infection while Treg cells of both genotypes released equivalent amounts of IL-10 (Fig. 5g and Supplementary Fig. 5g, h) suggesting fully functional Treg cells in RelB[ΔDC] mice.

Next, we sought to investigate whether the increased Treg cell numbers in RelB[ΔDC] mice prevented the induction of a protective Th2 response. Indeed, we found a ten-fold increase in the frequency of GATA3[+] Th2 cells in mLN and to a similar degree also in SI-LP of primary infected control mice whereas RelB[ΔDC] mice only showed a modest 2–3-fold increase (Fig. 5e and Supplementary Fig. 5i). After secondary *Hpb* infection, GATA3[+] Th2 cell frequencies even further increased in control animals while RelB[ΔDC] mice completely failed to further expand their Th2 effector cell population (Fig. 5e). This difference was not restricted to GATA3 expression but also manifested in a decreased release of IL-4 and IL-13 from restimulated T cells of RelB[ΔDC] mice post secondary *Hpb* infection (Fig. 5f and Supplementary Fig. 5j). Lastly, we also measured the relative distribution of cDC subsets after primary or secondary infection with *Hpb* to assess whether a different composition of cDCs post infection might contribute to an altered Th2 cell response. In general, infection with *Hpb* resulted in a reduced frequency of migratory cDCs (Fig. 5g), accompanied by a drastic accumulation of resident DCs in total cell numbers (Supplementary Fig. 5k). Among resident cDCs, the parasite infection resulted in an increase in cDC2s at the expense of cDC1s compared to non-infected animals (Fig. 5g and Supplementary Fig. 5l). Among migratory cDCs, we found a similar pattern with more cDC2s at the expense of cDC1s while the CD11b[+]CD103[+] migratory cDC frequency did not change after infection (Fig. 5g and Supplementary Fig. 5m). The overall distribution of cDCs in RelB[ΔDC] mice followed this pattern albeit differences in cDC distribution found in steady-state were grossly maintained after *Hpb* infection.

Altogether, these results demonstrate that RelB[ΔDC] mice fail to mount a protective Th2-driven immune response upon secondary *Hpb* infection resulting in elevated worm counts and a failure to trap parasites into submucosal granulomas. Elevated Treg cell counts in RelB[ΔDC] mice were maintained during infection and Treg cells of RelB[ΔDC] mice did not acquire Th2-like capacities despite elevated GATA3 expression and a strong Th2 cell-biased microenvironment.

## Dominant immune tolerance in RelB[ΔDC] mice is mediated via bystander Foxp3[+] Treg cell activity

Our results indicate that RelB controls several gene signatures associated with DC function in mLN at steady-state (Fig. 3). Therefore, we next investigated by another scRNAseq experiment whether these differences were maintained during an ongoing infection and/or whether additional differences exist in DC subsets during the priming or effector phase of *Hpb* infection. Interestingly, all RelB-dependent differences identified at steady-state were grossly maintained at day 4 post infection, when a Th2 immune response is initiated[47] or day 14 post infection, when *Hpb* may have induced robust Th2 immunity[48] (Fig. 6a). In line with flow cytometric observations (Fig. 5g), the proportion of cluster 4, composed of resident DC1s, was reduced upon *Hpb* infection (Fig. 6b). Moreover, we tested whether RelB-deficient DCs were still capable of upregulating chemokines (CCL17, CCL22) and pro-inflammatory cytokines (IL-6) upon stimulation. In line with previous results[49], granulocyte-macrophage colony-stimulatory factor (GM-CSF) was a potent inducer of CCL22 and CCL17 secretion from DCs while LPS and *Hpb* extract induced IL-6 release (Fig. 6c–e). Despite considerable variation, the differences in CCL17 and CCL22 observed in unstimulated samples were overall maintained suggesting a RelB-dependent dominant effect on the expression of these chemokines independent from the inflammatory environment.

As these and the previous results strongly suggest that elevated Treg cell frequencies mediate a dominant immune regulation over an

initial Th2 cell immune bias in RelB[ΔDC] mice, we finally aimed to prove the crucial role of Treg cells for preventing a protective Th2 cell-dominated immune response in RelB[ΔDC] mice. For this purpose, we repeated our secondary infection model in RelB[ΔDC] mice but attenuated the increased Treg cell frequency by antibody-mediated reduction of Treg cells during the first three weeks of the primary infection (Fig. 6f). Antibody treatment of RelB[ΔDC] mice did not completely deplete Foxp3[+] Treg cells in the peripheral blood but rather temporarily reduced their frequency to control levels during the primary infection and increased again after finishing the treatment at day 28 post primary infection (Supplementary Fig. 6a, b). This temporal reduction in Treg cell frequencies was sufficient to allow for Th2 effector cell expansion in the peripheral blood comparable to control animals (Fig. 6g and Supplementary Fig. 6c). Strikingly, isotype control antibody-treated RelB[ΔDC] mice preserved elevated worm counts when compared to non-treated control mice but anti-CD25 antibody-treated animals showed a tendency for reduced parasite trapping in granulomas and significantly lower worm counts after re-infection (Fig. 6h, i and Supplementary Fig. 6d). This effect manifested despite anti-CD25 antibody-treated RelB[ΔDC] mice had restored elevated Treg cell counts in mLN after secondary infection (Fig. 6j and Supplementary Fig. 6e). Surprisingly, MCPT-1 levels rather followed the genotype instead of the worm counts at day 63 after secondary *Hpb* infection (Supplementary Fig. 6f). As in unmanipulated RelB[ΔDC] mice (Fig. 5), we did not observe any consistent alterations in the composition of the DC subsets post secondary infection in anti-CD25 antibody-treated RelB[ΔDC] mice (Supplementary Fig. 6g-i).

Overall, these results demonstrate that systemically elevated Treg cell frequencies present in steady-state RelB[ΔDC] animals were sufficient to prevent an effective memory Th2 cell response. Attenuating the Treg cell frequencies to wildtype levels during primary *Hpb* infection was sufficient to restore a protective Th2 cell response after secondary *Hpb* infection. Thus, RelB deficiency in CD11c[+] cells causes a systemic increase in Foxp3[+] Treg cells that results in a dominant immune tolerance and limits immunity to *Hpb* infection despite an initial type 2 immune bias along the intestinal tract in RelB[ΔDC] mice. Among cDC2s in mLN, RelB particularly controls the transcriptional program of migratory cDCs including chemokines and co-stimulatory molecules all of which may contribute to the elevated Treg cell counts manifesting in steady-state RelB[ΔDC] animals.

## Discussion

Here, we provide direct evidence that deficiency in the alternative NF-κB member RelB in CD11c[+] cells results in a dominant tolerance in the intestinal tract mediated by the maintenance of a sustained increase of Foxp3[+] Treg cells. This dominant tolerance is also functional and maintained upon Th2-polarizing conditions such as helminth infection or allergic conditions. Complete knockout of RelB results in a strong T cell-driven autoimmune disease due to the important role of RelB for the maturation of thymic epithelial cells[20,50,51] and this autoimmune syndrome manifests with symptoms associated to atopic diseases such as atopic dermatitis or allergic responses in the lung[21,22]. These observations raised the question whether the conditional ablation of RelB in CD11c[+] cells, presumably DCs, would equally poise such animals to more severe reactions in type 2-associated immune responses not least because such animals intrinsically have higher IgE serum levels, elevated GATA3[+] T cell counts in the lamina propria and fail to induce normal levels of RORγt-expressing Treg cells known to counteract Th2 cell-dominated immune responses (Fig. 1 and refs. 4,27). In contrast to RORγt[+] Treg cells, RelB[ΔDC] mice possess increased frequencies of GATA3-expressing Treg cells which can become pathogenic and release Th2 cell-associated cytokines under certain conditions[42,52]. To our surprise, we did not find any hint for more severe anaphylactic reactions after induction of food allergy despite elevated amounts of allergen-specific IgE molecules in the serum of RelB[ΔDC] mice.

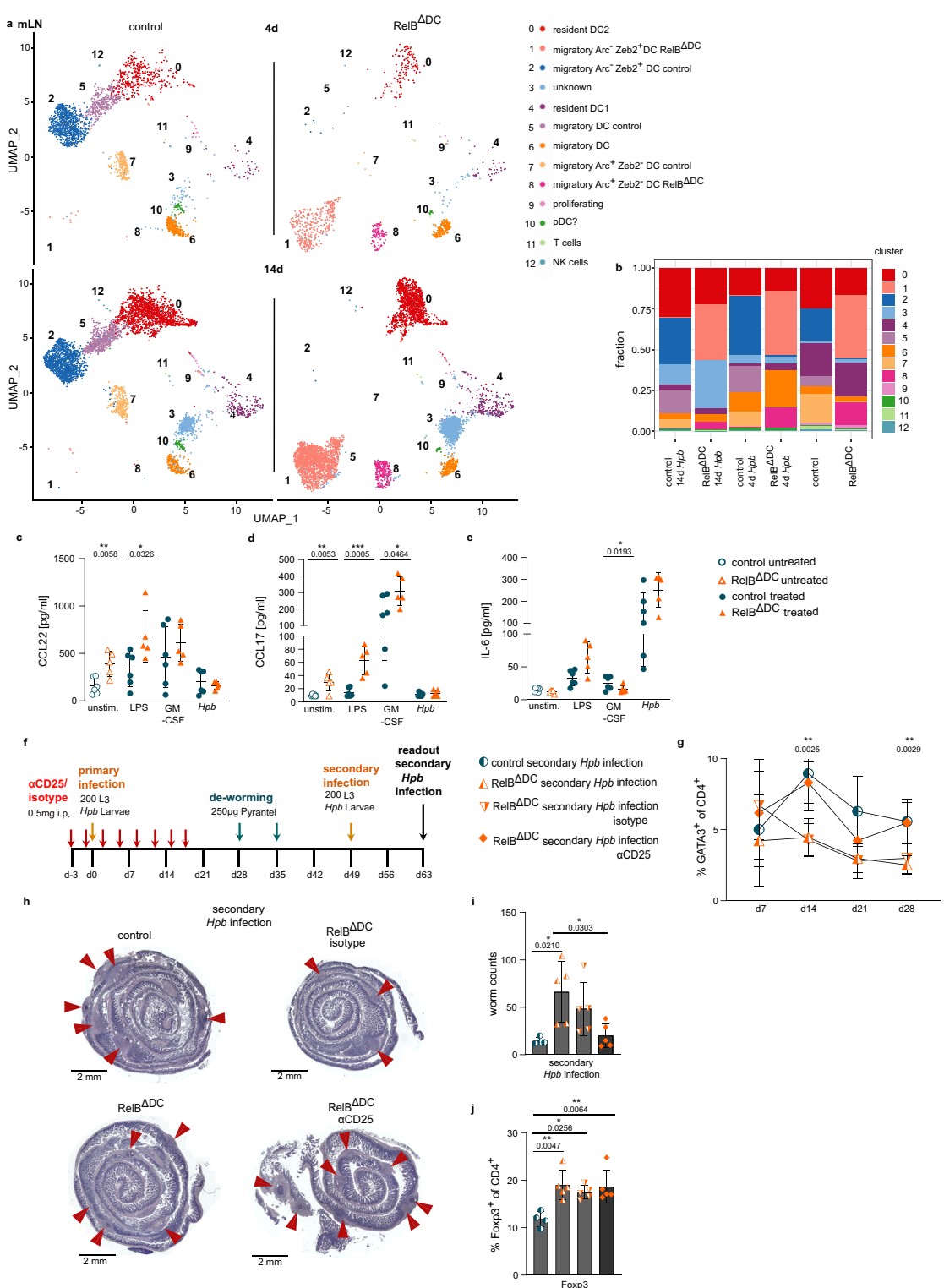

One possibility may be that Treg cells directly suppress the activity of sensitized type 2 innate lymphoid cells, basophils, mast cells, or macrophages. Alternatively, RelB-deficient DCs may either directly or indirectly favor the generation of allergen-specific antibody isotypes with an inhibitory function via Treg cell-dependent or -independent effects on the differentiation of allergen-specific follicular helper T cells.

Moreover, infection with the helminth *Hpb* did not result in exaggerated Th2 cell differentiation in RelB^ΔDC mice but conversely resulted in impaired protection against secondary *Hpb* infection due to attenuated Th2 cell expansion during primary and secondary infection. This attenuation was the result of Treg cell-mediated suppression because the temporal reduction of elevated Treg cell frequencies to control levels in RelB^ΔDC mice during primary *Hpb* infection

**Fig. 6 | Dominant immune tolerance results in reduced parasite clearance in RelB$^{\Delta DC}$ mice. a** UMAP projection of scRNAseq gene expression profiling from CD11c$^+$MHCII$^{hi}$ cells (among live/dead$^-$CD45$^+$ B220$^-$CD64$^-$) sorted from mesenteric lymph node (mLN) of control and RelB$^{\Delta DC}$ mice 4 days and 14 days after *Heligmosomoides polygyrus bakeri* (*Hpb*) infection. **b** Percentage distribution of DC clusters from the data of Fig. 2c (steady-state), day 4 and day 14 after *Hpb* infection (**a**). Levels of Ccl22 (**c**), Ccl17(**d**), and IL-6 (**e**) in supernatant of CD11c-enriched cells from mLN of control and RelB$^{\Delta DC}$ mice after 16 h stimulation with LPS, or GM-CSF or *Hpb* extract (control *n* = 6, RelB$^{\Delta DC}$ *n* = 5). **f** Experimental scheme of secondary *Hpb* combined with anti-CD25 antibody/isotype control antibody treatment. **g** Percentage of GATA3$^+$ of Foxp3$^-$ CD4$^+$ T cells in blood at indicated timepoints before and after *Hpb* infection. **h** Representative H&E staining of histology slides after secondary *Hpb* infection ± anti-CD25/isotype control antibody treatment.

Arrows indicate detected granulomas and are representative of 4 control, 5 RelB$^{\Delta DC}$ mice and 5 RelB$^{\Delta DC}$ ± anti-CD25/isotype-treated individual mice from one experiment. **i** Intestinal worm counts on day 63 after secondary *Hpb* infection of control and RelB$^{\Delta DC}$ mice ± anti-CD25/isotype control antibody treatment. **j** Quantification of Foxp3$^+$ CD4$^+$ Tregs in mLN after secondary *Hpb* infection ± anti-CD25/isotype control antibody treatment. Each dot represents an individual mouse and mean ± SD from one (**g**–**j**) or two independent experiments (**c**–**e**) is shown. control *n* = 4, RelB$^{\Delta DC}$ *n* = 5, RelB$^{\Delta DC}$ (αCD25) *n* = 5, RelB$^{\Delta DC}$ (isotype) *n* = 5. Statistical analysis was performed using two-tailed students *t*-test (**c**–**e**). For **g**, RelB$^{\Delta DC}$ mice were compared to anti-CD25 treated RelB$^{\Delta DC}$ mice with two-tailed students *t*-test. One-way ANOVA with Tukey correction for multiple comparison was used for (**i**, **j**). *P* value of <0.05 was considered statistically significant with **p* < 0.05, ***p* < 0.01, ****p* < 0.001, *****p* < 0.0001. Source data are provided as a Source Data file.

was sufficient to restore Th2 cell differentiation and control secondary infection (Fig. 6). It is well established that protection from primary and particularly secondary *Hpb* infection depends on the magnitude of the induced Th2 cell response[48,53,54]. Importantly, Treg cells from RelB$^{\Delta DC}$ mice did not show any signs of pathological reprogramming towards Th2 cells[42] as they secreted comparable amounts of IL-10 but not the Th2 cell-associated cytokines IL-4 or IL-13. Thus, it remains possible that the slightly elevated frequencies of GATA3$^+$ Th cells in naïve RelB$^{\Delta DC}$ mice might be derived from transdifferentiated GATA3-expressing tissue Treg cells. Such impaired protection by an expanded Treg cell population has already been observed in various parasite infections including *Hpb* and *Leishmania major* infections (ref. 55; reviewed in ref. 56). We extend this concept by showing that also indirect accumulation of Treg cells with a tissue Treg cell phenotype, driven by manipulated DCs, equally fulfills this function. Together with our earlier observation that RelB$^{\Delta DC}$ mice are also resistant to experimental autoimmune encephalomyelitis (EAE)[27], we propose here that the level of overall immune tolerance executed by the frequency of Foxp3$^+$ Treg cells is limited by RelB expression in DCs irrespective of the type of immune challenge.

How and in which DC subsets does RelB exert this function? Classical analysis of major DC subsets by flow cytometry did not reveal any difference in the frequency or number of lamina propria resident cDC1s or cDC2s. Initially, we expected that DCs collecting self- or microbial antigens in the lamina propria are primarily affected by the absence of RelB when considering the tissue phenotype of accumulated Tregs in RelB$^{\Delta DC}$ animals[27]. However, scRNAseq analysis allowed the detection of one DC cluster characterized by a specific gene expression profile (*Il22ra2, LTb, Lyz2, Il1rn*) distinct from cDC1 and cDC2s that was not detectable among DCs derived from RelB$^{\Delta DC}$ mice. The signature of this subset very closely resembles the expression profile of CIA-DCs[31]. Notably, CIA-DCs depend on the expression of LTβR expression and LTβ is a well-known trigger of the alternative NF-κB pathway[31,33]. These observations suggest that LTβ induces RelB-dependent pathways during the differentiation of CIA-DCs. Whether this LTβ is derived from type 3 innate lymphoid cells (ILC3s) or can also be derived from CIA-DCs themselves in an auto-/paracrine-manner may be a matter of timing during the differentiation of CIA-DCs. Besides the lack of CIA-DCs, we did not observe any difference in the distribution of DC subsets of the lamina propria nor a RelB-dependent gene expression profile. However, analysis of LtbR$^{\Delta DC}$ mice that lack CIA-DCs[31] did not show any effect on Treg cells excluding a causative role of CIA-DC deficiency for the increased Treg cell numbers in RelB$^{\Delta DC}$ mice.

In contrast, resident cDC2s from mLN were slightly reduced in their number thereby reverting the ratio of resident DC1s and resident DC2s in RelB$^{\Delta DC}$ relative to control mice. This difference was grossly maintained during food allergy and *Hpb* infection suggesting a stable effect of RelB on the absolute number of resident cDC2s in mLN. More interestingly, while no differences could be observed in the

transcriptional profile of resident DCs, scRNAseq analysis revealed fundamental differences in gene expression among migratory *Ccr7*$^+$ cDCs resulting in genotype-specific clustering. Cluster tree analysis and similarity analysis of hallmark genes still allowed us to compare individual clusters of control DCs (clusters 2 and 7, respectively) with their respective counterparts from RelB$^{\Delta DC}$ mice (clusters 1 and 8, respectively). Among these differentially expressed genes, RelB-deficient DCs expressed lower (*Cd200*) or higher (*Cd80, Cd86*) levels of co-stimulatory molecules and genes associated with cell migration (*Cd63*), while chemokines (*Ccl22, Ccl17*) were overexpressed. RelB-dependent expression of CD200, and CD63 as well as elevated production of CCL22 and CCL17 at steady-state conditions could also be confirmed on the protein level. Importantly, co-stimulatory molecules may interact not only with naïve or effector T cells but also with Foxp3$^+$ Treg cells and this interaction may act in a feed-forward loop rendering DCs more tolerogenic[57–59]. Furthermore, CCL22 has been shown to favor DC-Treg cell interaction[38] and also we found a selective advantage of Treg cells over non-Treg cells to interact with mLN DCs of RelB$^{\Delta DC}$ mice at an early time point after stimulation (Fig. 3). Given that T cells compete for rare self-antigen:MHC-II complexes, elevated CCL22 expression by RelB-deficient DCs may allow Treg cells improved access to survival and proliferation signals, resulting in enhanced Treg cell stimulation and expansion in RelB$^{\Delta DC}$ mice. Altered expression profiles of co-stimulatory signals of RelB-deficient DCs may contribute to additionally shape Treg cells to gain a 'tissue Treg cell' expression profile. Noteworthy, the described RelB signature partially resembles a tolerogenic signature of migratory DCs promoting resistance to checkpoint therapy in cancer, so-called 'mature DCs enriched in immunoregulatory molecules' (mregDCs)[19]. RelB might control a whole set of genes and pathways in migratory cDCs (Fig. 3) that in sum act as an 'immunological' rheostat in DCs to set the tolerogenic limit by restricting excessive Treg cell frequencies. It remains to be explored whether one determinant signal upstream of RelB may contribute to the regulation of the RelB rheostat. To our knowledge neither the inhibition or knockout of known non-canonical NF-κB drivers (CD40, LTβR, or TNF family members) in a DC context has resulted in a remarkable increase in Foxp3$^+$ Treg cells.

Maintenance of the Foxp3$^+$ Treg cell pool in the periphery requires constant stimulation of the TCR[60,61]. Furthermore, the acquirement of organ-specific Treg cell features originally derived from the thymus may require cognate interaction with self-antigens in secondary lymphoid organs expressed exclusively in the respective peripheral organ[62,63]. In line with this concept, precursors of GATA3$^+$, ST2$^+$ tissue Treg cells have been recently identified in the spleen and lymph nodes[29]. Similarly, we observed increased frequencies of such tissue Treg precursor cells in mLN of RelB$^{\Delta DC}$ mice (Supplementary Fig. 1i-k). Our work confirms the concept that cDCs, particularly those with a migratory phenotype[64], may constantly sample self-antigens in peripheral organs, migrate to secondary lymphoid organs and present such antigens to circulating thymus-derived Treg cells as shown for

adipose tissue Treg cells[63]. Importantly, cDCs may need to maintain an immature or tolerogenic status to avoid accidental activation of self-reactive non-Treg cells or induce trans-differentiation of thymic-derived Treg cells into effector T cells. In solid tumors, such tolerogenic DCs may represent a distinct subset that limits the efficacy of tumor eradication in some conditions[19]. As such mregDCs are characterized by the expression of RelB it is tempting to speculate whether also local effects on Treg cells in tumors or draining lymph nodes may contribute to resistance to checkpoint inhibition therapy. In the gut-draining mLN, the absence of non-canonical NF-κB signaling by RelB seems to enhance or imprint this feature in multiple ways, for instance by enhancing interaction time with Treg cells through the release of CCL22, by influencing the migratory behavior of cDCs and by regulating the expression of a unique set of costimulatory molecules.

A pro-tolerogenic phenotype in DCs devoid of non-canonical NF-κB signaling might explain very well the increase in Foxp3$^+$ Treg cells resulting in an overt immune tolerance in RelB$^{\Delta DC}$ mice. However, these results do not explain the reduction in microbiome-dependent RORγt$^+$Helios$^-$ peripherally induced Treg cells (pTreg cells) in these animals at steady-state and the lacking ability for the de novo induction of these cells[27] (Fig. 4i). This opposite effect is rather surprising considering the highly overlapping requirements for the differentiation of pTreg cells and circulating thymic derived Treg cells present in all peripheral organs. Two basic possibilities may be relevant in this regard: First, the absence of CIA-DCs in RelB$^{\Delta DC}$ mice may limit either the differentiation or the maintenance of pTreg cells in the SI-LP. In this case, CIA-DCs may provide unique pro-survival signals for local pTreg cells that cannot be compensated by other cDCs found in normal numbers at this site in RelB$^{\Delta DC}$ mice (Figs. 1 and 2). Second, another RelB-dependent CD11c$^+$ subset distinct from cDCs may be required for the differentiation and/or maintenance of pTreg cells within mLN. Indeed, three reports published during the preparation of this manuscript suggested that a RORγt-expressing antigen-presenting cell type highly abundant in early life supports the induction of microbiome-induced RORγt$^+$Helios$^-$ Treg cell population[65-67]. As our analysis focused on adult mice and on cDCs, we may have missed this population in our study. As RelB expression is crucial for mTEC development and maturation[51] and one RelB-expressing RORγt$^+$ APC shows a mTEC reminiscent transcriptional profile in mLN[65], we postulate that this cell type is equally targeted in RelB$^{\Delta DC}$ mice during early development. This could lead to the reduced induction of pTreg cells (Fig. 1c, d) maintained in adulthood and independent from our cDC analysis.

In summary, we propose that RelB expression in cDCs acts as a major and dominant rheostat to define the level of T cell-driven immune tolerance in the intestinal tract. RelB exerts this function by limiting the early recruitment of Treg cells to cDCs and by regulating a number of genes associated with migration and DC-Treg cell interaction, particularly in migratory DCs. The question if this effect depends solely on the alternative NF-κB pathway or interaction with the classical NF-κB pathway remains to be explored[68]. It is also of interest whether the upstream kinase NIK fulfills a similar function on immune tolerance as its deletion in CD11c$^+$ cells renders mice more susceptible to bacterial infection while at the same time protecting IL-10 knockout mice from colitis[69]. Whether the opposite effect, meaning boosting non-canonical NF-κB signaling in antigen-exposed cDCs, results in a lower tolerogenic set point and enables more efficient immune responses against tumors or chronic infections remains an interesting future avenue of investigation. This concept may be therapeutically exploited to fight chronic inflammation and autoimmune syndromes, especially, as it did not lead to an increased allergic risk in our hands.

## Methods

### Mice
Mice expressing a Cre recombinase under the control of the Itgax promoter/enhancer region (B6.Cg-Tg(Itgax-cre)1-1Reiz/J)[70] were purchased from The Jackson Laboratory (strain number 008068) and bred with mice carrying a loxP flanked exon 4 of the *Relb* gene (RelB$^{fl/fl}$)[51] to achieve deletion of RelB in DCs (called RelB$^{\Delta DC}$ throughout the manuscript). LtbR$^{\Delta DC}$ mice were generated and used as described in ref. 31. Age- and sex-matched male and female mice at 8–14 weeks of age (unless the duration of the experiment required a longer lifespan) were used and Cre$^-$ littermates were chosen as control animals whenever possible. Animals were tested for *Relb* germline deletion when necessary. TCR-transgenic OT-II animals (B6.Cg-Tg(TcraTcab)425CBn/Crl) were purchased from Charles River Laboratories (strain code 643) and bred at the LMU University Hospital. All animals were kept under specific pathogen-free conditions in individually ventilated cages. All interventions were performed in accordance with the European Convention for Animal Care and Use of Laboratory Animals and were approved by the local ethics committee and appropriate government authorities (Regierung von Oberbayern, license numbers ROB55.2-2532.Vet_02-15-5, ROB55.2-2532.Vet_02-17-222 and ROB55.2-2532.Vet_02-21-48).

### Isolation of Immune cells
**Steady-state lamina propria of the small intestine.** For preparation of single-cell suspensions from SI-LP the small intestine was removed, placed in ice-cold DPBS (Gibco) and Peyer's patches were removed. Intestines were cut open longitudinally, washed with ice-cold PBS, and cut into 0.5–1 cm pieces. For mucus removal, small intestine pieces were incubated in RPMI 1640 (Gibco) containing 25 mM HEPES (Gibco), 5 mM EDTA (Invitrogen), 3% FCS (Sigma Aldrich), and 0.145 mg/ml DTT (Sigma) for 20 min at 37 °C, 80 rpm. Thereafter, tissues were washed three times with RPMI1640 containing 25 mM HEPES and 2 mM EDTA and minced with sharp scissors into 1–2 mm pieces. To generate single-cell suspensions, minced pieces were digested for 30 min by shaking at 100 rpm at 37 °C in RPMI 1640 containing 25 mM HEPES, 200 iU/ml Collagenase IV (Worthington), and 0.2 mg/ml DNaseI (Sigma). Tissue pieces were then pipetted up and down several times and cell suspensions were collected. One more cycle of digestion with fresh medium was performed and cell suspensions were filtered through a 100 μm cell strainer (Corning). Cell suspension was centrifuged at 450 g, for 5 min at 4 °C, cell pellets were resuspended in 40% Percoll (Cytiva) and layered onto an 80% Percoll solution. The Percoll gradient was centrifuged at 1600 g at room temperature for 15 min. The interlayer containing lamina propria mononuclear cells was collected in FACS buffer (DPBS containing 1% FCS, 25 mM HEPES, and 2.5 mM EDTA). This single-cell suspension was then used for downstream analysis.

**Lamina propria of the small intestine after Hpb infection.** Isolation of lamina propria cells from *Hpb*-infected mice was performed using an alternative cell isolation protocol, adjusted to the high inflammatory status and mucus production[71]. Small intestinal tissues were harvested as described before, but for mucus removal, tissue pieces were incubated in HBSS (Gibco) containing 2 mM EDTA for 10 min at 37 °C with shaking at 200 rpm, and afterward pulse vortexed three times for 5 s at 2500 rpm. The supernatant was discarded and fresh HBSS containing 2 mM EDTA was added. This process was repeated four times. For digestion, washed small intestinal pieces were minced and incubated two times with RPMI 1640 containing 20% FCS, 0.2 mg/ml DNaseI, and 1 mg/ml Collagenase A (Roche) for 15 min at 37 °C and shaking at 200 rpm. Cell suspensions were filtered through a 100 μm strainer, collected in FACS buffer, and stored on ice. Percoll gradient was performed as described for the steady-state protocol.

**Lymphoid organs.** To generate single-cell suspensions from lymphoid organs, spleen and indicated lymph nodes were collected in RPMI1640 containing 25 mM HEPES and 5% FCS, and tissues were minced into 1–2 mm pieces. Tissue pieces were digested by incubation in RPMI

1640 containing 25 mM HEPES, 200 iU/ml Collagenase IV (Worthington), and 0.2 mg/ml DNAseI (Sigma) for 30 min at 37 °C, shaking at 100 rpm. Digestion was skipped for determination of CD117 expression levels on mLN DCs to preserve antigen integrity (Fig. 3i, j). Cell suspensions were filtered through a 70 μm cell strainer, collected in FACS buffer and centrifuged at 450 g for 5 min at 4 °C. For the spleen, erythrocytes were removed by incubating cells in 2 ml of ACK lysis buffer (0.15 M ammonium chloride (Sigma), 10 mM potassium hydrogen carbonate (Merck), 1 mM EDTA-disodium, pH-adjusted to 7.3) for 2 min, before samples were washed and resuspended with respective buffers for further analysis.

### Flow cytometry
**Extracellular staining.** Cell numbers were routinely counted with an Accuri C6 Flow Cytometer (BD). Single-cell suspensions were first incubated for 10 min at 4 °C with an Fc blocking antibody in FACS buffer (CD16/CD32, BD, catalog number 553142, dilution 1:300), followed by 20 min incubation at 4 °C with the indicated antibodies in FACS buffer. 7-Aminoactinomycin D (7-AAD) (Enzo) or fixable Zombie Aqua (BioLegend) was used for live/dead cell discrimination according to the manufacturer's instructions.

**Intracellular staining of transcription factors.** Intracellular staining of transcription factors was performed with a Foxp3 fixation/permeabilization kit (ThermoFisher) according to the manufacturer's instructions. Fixed cells were intracellularly stained with the indicated antibodies for 60 min at room temperature.

**Intracellular staining of cytokines.** For staining of intracellular cytokines, $3 \times 10^6$ cells in single-cell suspension were seeded in U-bottom 96-well plates (Sarstedt). Cells were incubated in complete RPMI (RPMI 1640, 2 mM L-Glutamine (Gibco), 10,000 U Penicillin G and 10 mg/ml Streptomycin (Gibco), 10% FCS, 50 μM β-mercaptoethanol (Sigma)) containing 200 ng/ml PMA (Sigma) and 1 μg/ml Ionomycin (Cayman) for two hours at 37 °C, 5% $CO_2$. Afterwards, 5 μg/ml Brefeldin A (Sigma) was added, and cells were incubated for an additional two hours. Subsequently, cells were harvested, incubated with Fc block, and extracellular staining was performed by incubating samples with indicated antibodies for 20 min at 4 °C. For the fixation of samples, the Cytofix/Cytoperm Kit (BD) was used according to the manufacturer's protocol. Intracellular staining was performed overnight at 4 °C. Fluorescence minus one (FMO) controls were performed to control the quality of cytokine staining.

All flow cytometric experiments were performed on an LSR Fortessa (BD) and FlowJo (version 10) (Tree Star) software was used for data analysis. Gating strategies for T cell effector subsets, regulatory T cells, and dendritic cell populations in all organs are shown in Supplementary Fig. 7.

**Antibodies.** The following antibodies were purchased from BioLegend: anti-mouse CD103-PE (clone 2E7, catalog number 121406, dilution 1:100), CD117-PerCP/Cy5.5 (clone 2B8, catalog number 105824, dilution 1:100), CD11c-BV786 (clone N418, catalog number 117335, dilution 1:200), CD172α-PE-Cy7 (clone P84, catalog number 144008, dilution 1:100), CD200-APC (clone OX-90, catalog number 123180, dilution 1:100), CD3ε-AF700 (clone 17A2, catalog number 100216, dilution 1:100), CD4-BV786 (clone GK1.5, catalog number 100453, dilution 1:400), CD45.2-Pacific Blue (clone 104, catalog number 109820, dilution 1:200), CD45R-AF488 (clone RA3-6B2, catalog number 103225, dilution 1:200), CD63-APC (clone NVG-2, catalog number 143906, dilution 1:100), CD64-APC (clone X54-5/7.1, catalog number 139306, dilution 1:100), KLRG1-BV421 (clone 2F1, catalog number 138414, dilution 1:100), MHC II-AF700 (clone M5/114.15.2, catalog number 107622, dilution 1:300) and XCR1-BV650 (clone ZET, catalog

number 148220, dilution 1:200). The following antibodies were purchased from eBioscience: Anti-mouse CD11b-APC-e780 (clone M1/70, catalog number 47-0112-82, dilution 1:100), CD44-APC-eF780 (clone IM7, catalog number 47-0441-80, dilution 1:200), CD45-APC-eF780 (clone 30-F11, catalog number 47-0451-82, dilution 1:100), CD45-FITC (clone 30-F11, catalog number 11-0451-82, dilution 1:500), CD8α-AF700 (clone 53-6.7, catalog number 56-0081-82, dilution 1:100), Foxp3-PerCP-Cy5.5 (clone FJK-16s, catalog number 45-5773-82, dilution 1:100), GATA3-eF660 (clone TWAJ, catalog number 50-9966-42, dilution 1:20), IL-13-PE (clone eBio13A, catalog number 12-7133-82, dilution 1:100) and RORγt-PE (clone AFKJS-9, catalog number 12-6988-82, dilution 1:100) and ST2(IL33R)-Biotin (clone RMST2-2, catalog number 13-9335-82, dilution 1:100). From BD, anti-mouse CD3ε-FITC (clone 145-2C11, catalog number 553062, dilution 1:100), CD4-AF700 (clone RM4-5, catalog number 557956, dilution 1:400), IL-10-BV711 (clone JES5-16E3, catalog number 564081, dilution 1:100) and IL-4-BV421 (clone 11B11, catalog number 562915, dilution 1:100) were used and CD8α-APCeF780 (clone 5H10, catalog number 47-4321-82, dilution 1:600) was purchased from Invitrogen. Biotinylated anti-mouse Plet1 was obtained from C. Ruedl[32] and used in combination with a streptavidin PE-Cy7 conjugate (BioLegend, catalog number 405206, dilution 1:1000).

### Food allergy model
To elicit food allergy in control and RelB$^{ΔDC}$ mice, an OVA-driven model analog to an established model of peanut allergy was used[43]. Briefly, animals were intragastrically gavaged with 10 μg cholera toxin (Biological Lab) and 5 mg chicken albumin egg grade III (OVA, Sigma) in DPBS on days 0, 2, 7, 14, 21, and 28. For control groups, mice were gavaged with 10 μg cholera toxin with DPBS only. On day 35, all mice were challenged by intravenous injection with 25 μg OVA EndoFit (InvivoGen) in DPBS, and temperature was assessed by rectal measurement every 5 min. The maximal temperature drop was calculated by subtracting the lowest temperature being measured from steady-state temperature (determined before challenge) for each mouse individually. Blood samples for analysis of MCTP-1 levels were taken 90 min after challenge, organs and blood samples for OVA-specific IgE levels were collected on day 36.

### Infection with *Heligmosomoides polygyrus bakeri*
**Primary infection.** Control and RelB$^{ΔDC}$ mice were intragastrically infected with 200 L3 stage larvae of *Hpb* in DPBS on day 0. At day 21 post infection, the small intestine was isolated and placed in a petri dish containing ice-cold DPBS. Peyer's patches were removed, and tissue was cut open longitudinally. The small intestinal content including worms was thoroughly washed into the petri dish and *Hpb* larvae were counted in a blinded manner. Immune cell isolation from indicated organs was done as described in the isolation section.

**Secondary infection.** For secondary infections, mice were de-wormed by intragastric gavage with 250 μg Pyrantel (InfectoPharm) in 200 μl DPBS on day 28 and day 35 after primary infection[72]. On day 49 post infection, mice were re-infected by intragastric gavage with 200 L3 larvae of *Hpb* in DPBS. On day 63, mice were sacrificed by cervical dislocation, worms were counted as described for primary infection. 7 cm of the proximal end of small intestine were used for preparation of Swiss rolls and fixed in 4% Formaldehyde (Honeywell) for 24 h at room temperature. Immune cells from indicated organs were isolated as described in isolation section.

**Treg cell reduction by anti-CD25 antibody treatment during primary Hpb infection.** For systemic reduction of regulatory T cell numbers during *Hpb* infection, control and RelB$^{ΔDC}$ mice were injected intraperitoneally with 0.5 mg of anti-CD25 antibody (clone PC-61.5.3;

BioXCell, catalog number BP0012) or an isotype control antibody (clone HRPN; BioXcell, catalog number BP0088), in 200 μl DPBS on days −3 and −1 before mice were infected with *Hpb* on day 0 as described for primary infection. Treatment with anti-CD25 or isotype control antibodies was continued on days 3, 6, 10, 13, 17 and 20. To control the efficiency of regulatory T cell depletion during treatment, blood was collected by retrobulbar bleeding into heparin microvettes (Sarstedt). After ACK lysis, cells were stained extra- and intracellularly for flow cytometric analysis as described in previous sections.

## Histological analysis by H&E staining
Proximal 7 cm of the small intestine was fixed in 4% Formaldehyde (Honywell) as Swiss rolls for 24 h, followed by paraffin embedding (Leica). 4 μm tissue sections were stained with hematoxylin and eosin (H&E) for histological analysis. Pictures were acquired on an AxioScan 7 (Zeiss) with 20× magnification and granuloma numbers were counted per slide.

## Fluorescence-activated cell sorting of dendritic cells
Single-cell suspensions from the SI-LP and mLN were generated as described above. For mLN, a DC enrichment step was performed before cDC sorting using a CD11c enrichment kit (Miltenyi) according to the manufacturer's protocol. Enriched DCs were stained extracellularly and 7AAD was added directly before sorting. Cell sorting was performed with a FACSAria Fusion cell sorter (BD) and populations were sorted according to the gating strategy in Supplementary Fig 7. Purity check was routinely performed with each sample and sort purity typically reached >99%. For single-cell RNA-sequencing analysis, live CD45$^+$CD64$^-$,B220$^-$ CD11c$^{hi}$MHC-II$^+$ cells from three (steady-state) and two (after *Hpb* infection) mice for each group were pooled before sorting to prevent individual bias due to cell isolation in individual animals.

## Single-cell RNA-sequencing
Gene expression libraries for single-cell RNA-sequencing were prepared by using Chromium Next GEM Single Cell 3' GEM Kit v3.1 (10X genomics) and the 10× Chromium Controller (10× genomics) following the manufacturer's protocol (CG000206 RevD). Dual Index Kit TT Set A (10× genomics) has been used for library preparation and samples were sequenced in a paired-end run (R1/28, i7/10, i5/10, R2/90) on a NovaSeq 6000 platform using SP v1.5 (100 cycles, Illumina) for steady-state and S2 v1.5 (100 cycles, Illumina) for *Hpb*-infected samples.

## Single-cell RNA-sequencing analysis
We used cellranger (version 7.0.1) to align the reads to the mouse genome (10× genomics reference build MM10 2020 A) and obtain per-gene read counts. Subsequent data processing was performed in R using Seurat (version 4.1.1) with default parameters if not indicated otherwise. After merging the data, we scaled the data (normalization.method = 'LogNormalize', scale.factor = 10,000), detected variable features (selection.method = 'vst', nfeatures=2000), and scaled the data (vars.to.regress = c('nCount_RNA')). We then applied quality control filters on cells with the following criteria: a) more than 200 genes detected b) less than 20% mitochondrial gene reads c) more than 5% ribosomal protein genes reads d) less than 20% hemoglobin genes reads e) singlets as determined by doubletFinder (version 2.0.3, pK = 0.09, PCs =1:10). Only genes detected in at least 4 cells were kept. The resulting dataset consisted of 23,219 cells and 29,846 genes.

## cDC–T cell coculture assay
Treg and non-Treg cells were isolated from OT-II mice by magnetic cell sorting (CD4+ CD25+ Regulatory T Cell Isolation Kit, Miltenyi) and stained with CellTrace™ Far Red (red) and CellTrace™ CFSE (green; both Invitrogen), respectively. $5 \times 10^4$ of the labeled Treg and non-Treg cells each were cultured in a 96-well flat bottom plate, together with $5 \times 10^3$ DCs, isolated from mLN with a CD11c enrichment kit (Miltenyi) of control and RelB$^{\Delta DC}$ mice, respectively. After addition of 1 μg/ml OVA$_{323-339}$ peptide cells were incubated at 37 °C with 5% CO$_2$ and monitored with the IncuCyte® Live Cell Analysis System (Sartorius). The ratio of Treg to non-Treg cells per DC cluster was determined after 4.5 and after 6 h of culturing in a minimum of 10 clusters per biological replicate and time point by three blinded investigators.

## Stimulation of dendritic cells
Steady-state expression of cytokines and chemokines of cDCs isolated from mLN of control and RelB$^{\Delta DC}$ mice was analyzed by ex vivo stimulation. Briefly, single-cell suspensions from digested mLN were prepared, and CD11c$^+$ cells were enriched with the CD11c enrichment kit (BioLegend) according to the manufacturer's protocol. $5 \times 10^5$ enriched (CD11c$^+$) cells were seeded into 96-well-V-bottom plate and incubated in 100 μl of complete medium (RPMI 1640, 2 mM L-Glutamine (Gibco), 10,000 U Penicillin G and 10 mg/ml Streptomycin (Gibco), 10% FCS, 50 μM β-mercaptoethanol (Sigma)) ± stimulus at 37 °C for 16 h. Cells were either left unstimulated (no stimulus added), incubated with either 100 ng/ml LPS (InvivoGen), or 100 ng/ml GM-CSF (PeproTech) or 20 ng/ml *Hpb* extract.

## Quantification of cytokines and chemokines in DC supernatant
Cytokines and chemokines in DC supernatant were measured with a LEGENDplex mouse M2 macrophage panel (6-plex, BioLegend) according to the manufacturer's instructions.

## Enzyme-linked immunosorbent assay (ELISA) of MCPT-1 and OVA-specific IgE
To determine serum MCPT-1 and OVA-specific IgE antibody levels, retrobulbar blood was collected into serum microvettes (Sarstedt). After 20 min incubation at room temperature, serum was centrifuged at 10,000*g* at 4 °C for 10 min and samples were stored at −80 °C until analysis. MCPT-1 serum levels were measured with the mouse MCPT-1 uncoated ELISA kit (Invitrogen) according to the manufacturer's protocol. For the determination of OVA-IgE, the mouse OVA-specific IgE kit (BioLegend) was used according to the manufacturer's instructions.

## Statistics
For statistical analysis, GraphPad Prism (version 9.5.1) was used (excluding scRNAseq data). If not stated differently, each dot represents an individual mouse and mean ± SD is shown. Statistical analysis was performed using two-tailed students *t*-test or one-way ANOVA with Tukey correction for multiple comparisons. A *P* value of <0.05 was considered statistically significant. *$p < 0.05$, **$p < 0.01$, ***$p < 0.001$, ****$p < 0.0001$, n.s. not significant.

## Reporting summary
Further information on research design is available in the Nature Portfolio Reporting Summary linked to this article.

# Data availability
The authors declare that the data supporting the findings of this study are available within the paper and its Supplementary Files. Raw data are available on request. Raw datasets related to single-cell sequencing experiments of this study have been deposited and made publicly available in the Gene Expression Omnibus under the accession number GSE243483 Source data are provided with this paper in the Source Data file.

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

## Acknowledgements

We thank Johanna Grosch, Benjamin Schnautz and Anela Arifović for technical assistance. We also thank the Esser-von-Bieren lab for support with *Hpb* infections. This work was funded by grants from the European Research Council (ERC Starting grant, project number 716718) and the German Research Foundation (Project Number OH 282/1-2 within FOR2599, Project Number 490846870–TRR355/1 TPA05 and TPB02 and Project Number 395357507–CRC1371 TP07) to C.O. T.B. was supported by the German Research Foundation CRC1371 project P06 project-ID 435874434, RTG2668 project A2 and project B3 and by BI 696/14-1 project-ID 527318848. A.D. was supported by grants from the European Research Council (ERC Advanced grant, project number 101055309) and the German Research Foundation (Project Number Di764/10-2 within FOR2599); Project Number Di764/9-2 within SPP1937 and Project Number 427826188 within SFB1444).

## Author contributions

A.L.G. performed most experiments, data analysis and data interpretation. D.K., A.P., L.H., A.K., L.K., N.W. and D.A. contributed to experiments and data analysis. M.H. and T.B. helped with cell sorting. CR provided important reagents. M.R. and A.D. provided critical mouse strains. D.V. provided *Hpb* larvae. C.S.W. provided critical input, helped with paper writing and supported data interpretation. T.S. performed bioinformatic analysis. M.P. helped with experiments, data interpretation and paper curation. C.O. supervised the study, performed data interpretation, and acquired funding. A.L.G. and C.O. wrote the paper with input from all co-authors. All authors read and approved the paper.

## Funding

## Competing interests

The authors declare no competing interests.

## Additional information

[1]Center of Allergy and Environment (ZAUM), Technical University and Helmholtz Center Munich, Munich, Germany. [2]Division of Clinical Pharmacology, LMU University Hospital, LMU, Munich, Germany. [3]Department of Dermatology and Allergy Biederstein, School of Medicine and Health, Technical University of Munich, Munich, Germany. [4]School of Biological Sciences, Nanyang Technological University Singapore, Singapore, Singapore. [5]Leibniz Institute on Aging, Fritz Lipmann Institute, 07745 Jena, Germany. [6]Department of Medicine II, LMU University Hospital, LMU, Munich, Germany. [7]Laboratory of Innate Immunity, Institute of Microbiology, Infectious Diseases and Immunology, Charité-Universitätsmedizin Berlin, 12203 Berlin, Germany. [8]Mucosal and Developmental Immunology, Deutsches Rheuma-Forschungszentrum, an Institute of the Leibniz Association, 10117 Berlin, Germany. [9]Department of Infection Biology, University Hospital Erlangen and Friedrich-Alexander University Erlangen-Nuremberg (FAU), Erlangen 91054, Germany. [10]Member of the German Center of Lung Research (DZL), Partner Site Munich, Munich, Germany. [11]Bioinformatics Core Unit, Biomedical Center, Ludwig-Maximilians-University, 82152 Planegg, Germany. [12]Present address: Immatics Biotechnologies GmbH, Paul-Ehrlich-Str. 15, 72076 Tuebingen, Germany.
✉e-mail: caspar.ohnmacht@helmholtz-munich.de

