## [Transparent Peer Review file · Nature Communications]

Dominant immune tolerance in the intestinal tract imposed by RelB-dependent migratory dendritic cells regulates protective type 2 immunity

Corresponding Author: Dr Caspar Ohnmacht

Version 0:

Reviewer comments:

Reviewer #1

(Remarks to the Author)

The manuscript by Geiselhöringer et al examines the role of RelB in the development and functions of several intestinal cDC subsets by using *ItgaxCre x RelB^{fl/fl}* (RelB Δ DC) mice. RelB Δ DC mice showed an increase in total Foxp3⁺ Treg cells, including GATA3⁺Foxp3⁺ Treg cells, in mesenteric lymph nodes (mLN) and the lamina propria of the small intestine (SI-LP). The development of CIA-DCs seems to depend on RelB as CIA-DCs were completely absent in SI-LP of RelB Δ DC mice. Although RelB Δ DC mice harbored almost normal frequency and cell numbers of cDC subsets in mLN and SI-LP, migratory cDCs in mLN of RelB Δ DC mice exhibited the gene signatures favoring DC-Treg cell interaction by affecting the expression of chemokines, migration behavior, co-stimulatory molecule, and tolerance-related integrin expression. RelB Δ DC mice exhibited the mild anaphylactic reaction despite of the elevated allergic immune responses in a model of food allergy. While RelB Δ DC mice failed to mount a protective Th2 immune response against secondary Hpb infection, the depletion of Foxp3⁺ Treg cells by anti-CD25 mAb restored this response in RelB Δ DC mice. Of the results in this manuscript, the finding that RelB controls intestinal DCs is perhaps the novel and interesting, however mechanistic evaluation or a direct link between this and Foxp3⁺ Treg cells in type 2 immunity is lacking. Therefore, the manuscript, in its current state, raises several questions.

General:

The authors showed the impact of the deficiency of RelB on the development and functions of several intestinal cDC subsets. Can the deficiency of RelB affect the development and functions of cDC subsets in other tissues? How RelB regulates the development of CIA-DCs from cDC progenitors? RelB Δ DC mice displayed the enhanced proportion of Foxp3⁺ Treg cells, while they lacked CIA-DCs. It is not clear to this reviewer how CIA-DCs controls Foxp3⁺ Treg cells. The authors should comprehensively characterize this intestinal cDC subset, including immunogenic to generate Th cells and suppress Treg cells or tolerogenic to generate Treg cells, in a model of type2 immunity. Although the authors analyzed the gene expression profiles of migratory cDCs in mLN in RelB Δ DC mice by scRNAseq, it remains unclear how the deficiency of RelB regulates their functions leading to the enhanced proportion of Foxp3⁺ Treg cells to suppress type 2 immunity, including anaphylactic reaction in food allergy and Hpb infection in mice. In the Results, several speculations were found in each paragraph without any direct evidences, including Line 188-192, Line 233-235, Line 264-267, and Line 363-366. In addition, several misprints/abbreviations/syntax problems were detected, including lines 51, 60, 146, 146-147, and 166, among others.

Major comments:

1) Fig.1: It remains unclear why GATA3-expressing Treg and T helper cells increased in mLNs and SI-LP of RelB Δ DC mice. The authors also should provide flow cytometry plots, frequencies, and, total numbers in each panel to appreciate the changes in the composition of immune cells between RelB Δ DC mice and control mice.

2) Fig.2: CIA-DCs were completely absent in SI-LP of RelB Δ DC mice. How RelB controls the development of CIA-DCs? The authors indicate the connection of LT β signaling and CIA-DCs. Thus, the authors should address this by examining similar responses in *ItgaxCre x Lt β receptor fl/fl* and/or *Lt β receptor KO* mice. The authors indicated that the migratory cDC subsets in mLN of RelB Δ DC mice exhibited the gene signatures favoring DC-Treg cell interaction. However, they did not show the functional differences between control cDCs and RelB-deficient cDCs. The authors should directly address the capacity of each intestinal cDCs in both control mice and RelB Δ DC mice to generate Foxp3⁺ Treg cells from Foxp3⁻ T cells

in the presence or absence of active and latent form of TGF- β in vitro. This would provide more informative data than gene expression profiles alone. Total numbers should be provided in panels of Fig2h and Fig2i.

3) Fig.3: It is unclear to this reviewer why RelB Δ DC mice did not exhibit similar anaphylactic reaction to control mice, while they also manifested the elevated allergic immune responses in a model of food allergy. Can the elevated proportion of Foxp3⁺ Treg cells contribute to inhibit the anaphylactic reaction in RelB Δ DC mice? If so, how Treg cells suppress the pathogenesis of anaphylactic reaction without inhibiting allergic immune responses? In addition to frequencies, the authors also should provide flow cytometry plots and total numbers in panels of Fig.3d-j.

4) Fig.4: RelB Δ DC mice failed to mount a protective Th2-driven immune response upon secondary Hpb infection. Furthermore, RelB Δ DC mice showed the elevated number of Treg cells during infection and their Treg cells did not acquire Th2-like capacities despite elevated GATA3 expression and a strong Th2-biased microenvironment. However, the causation is not established with the presented study. In panels of Fig.4h-j, the authors should provide flow cytometry plots and total numbers.

5) Fig.5: The authors used anti-CD25 mAb to deplete Foxp3⁺ Treg cells in RelB Δ DC mice. However, the use of this mAb seems to be problematic since it could deplete Foxp3⁺ Treg cells as well as effector T cells during the immune responses as described elsewhere in the published reports. To precisely analyze the role of Foxp3⁺ Treg cells, it could be better to use RelB Δ DC mice x DREG (depletion of regulatory T cells) mice (J Exp Med. 2007;204:57-63). Although treatment of RelB Δ DC mice with anti-CD25 mAb caused an insufficient depletion of Foxp3⁺ Treg cells, but it restored a protective Th2 response after secondary Hpb infection. They indicated "Thus, RelB deficiency in CD11c⁺ cells causes a systemic increase in Foxp3⁺ Treg cells that result in a dominant immune tolerance and limits immunity to Hpb infection ...". This claim is not supported by the provided data and there is also a lack of cited evidence to back this up. For example, the suppressive activity of Foxp3⁺ Treg cells from RelB Δ DC mice against the activation of Th2 cells from Hpb-infected control mice in vitro. Alternatively, the suppressive effect of adoptive transfer with Foxp3⁺ Treg cells from RelB Δ DC mice against the protective Th2 immune response against secondary Hpb infection in control mice. To prove the crucial role of dominant immune tolerance mediated via bystander Foxp3⁺ Treg cell activity for preventing a protective Th2 immune response against Hpb infection in RelB Δ DC mice, the authors should consider performing such experiments. In addition to frequencies, the authors should provide flow cytometry plots and total numbers in panels of Fig.5m.

Minor comments:

- 1) Line 120-141: The description in the paragraph is too long and redundant with Results.
- 2) Line 51: Abbreviate type 2 T helper cells (Th2) cells. Please check Th in the manuscript throughout.
- 3) Line 60: Correct T helper cell to Th cell.
- 4) Line 146: Correct T helper cells to Th cells.
- 5) Line 146-147: Abbreviate mesenteric lymph nodes (mLN) and the lamina propria of the small intestine (SI-LP). In Figures, authors indicated mLN and SI-LP. They are abbreviated in Line 944-945. Please check the manuscript throughout.
- 6) Line 166: migratory DCs are CD11c^{int}MHC-II^{hi}. See Line 959.
- 7) Line 238: Data for GATA3⁺ Th2 cells is indicated in Fig.1e.
- 8) Line 415: DC subset surrounding cryptopatches and isolated lymphoid follicles (CIA-DCs) is abbreviated before (L183).
- 9) Line 432-433: RelB-deficient DCs expressed lower levels of co-stimulatory molecules (Cd80, Cd86, Cd200)... Fig.2f shows RelB-deficient DCs expressed higher or lower levels of CD80 and CD86 or CD200.
- 10) Line 443: Indicate the words for the abbreviation of mregDCs.
- 11) Line 944-945: mesenteric lymph nodes (mLN) and the lamina propria of the small intestine (SI-LP).
- 12) Line 952: correct T helper cells to Th cells. In manuscript, the authors used Th2 cells and Th17 cells elsewhere.
- 13) Line 957: DPDC2s (CD103⁺CD11b⁺) and SPDC2s (CD103⁻CD11b⁺). Indicate the words for the abbreviation of DP (double positive) and SP (single positive).
- 14) Line 974, 977, Correct C11c⁺MHCII^{high} DCs and CD11c⁺MHCII^{hi} DCs to C11c⁺MHCII^{high} DCs and CD11c⁺MHCII^{hi} DCs.
- 15) Line 983-985, Correct e, f to h, i.
- 16) Line 993: ovalbumin (OVA) is abbreviated before (Line 244).
- 17) Line 1001: expression in Foxp3⁺ CD4⁺ Tregs (i) in mLN. Fig.3h, I indicate the date for SI-LP. Correct mLN to SI-LP.
- 18) Line 1020: correct T helper cells to Th cells.
- 19) Line 1023: k Quantification of percentage of resident... Fig.4j indicates the data. Correct K to j.
- 20) Line 1031: Correct comparison (f, h, k) to comparison (f, h, j).
- 21) Check the Figure legends of supplementary figures as described above.

Reviewer #2

(Remarks to the Author)

The authors investigated the effect of specific RelB deficiency in CD11c⁺ DCs on regulatory T cells in the intestinal tract and their changes after induction of type 2 immunity by helminth or allergy induction.

They find that in mice with DC-specific RelB-deficiency the overall frequency and numbers of both resident and migratory DC are not affected in the lamina propria unlike in mesenteric lymph nodes, where resident cDC2 were decreased. This indicates that cDC2 development is affected RelB-deficiency as has been reported before (<https://doi.org/10.1002/eji.201747332>). No changes were observed for migratory DC subsets in the mesenteric lymph nodes (Fig. 1j). Here, the authors should show whether this is a result of the gating strategy for cDC2 by only including CD11b⁺ cells. What are then the CD11b⁻ cells in Fig. 1h? Is CD11b simply downregulated on their SPDC2 subset?

The data further indicate that RelB deficiency in DCs promoted GATA3⁺ Foxp3⁺ tTreg and GATA3⁺ non-Treg expansion in

the lamina propria and mesenteric lymph nodes. Thus, the T cell expansion did not correlate with an increased frequency of resident or migratory DC subsets in the lamina propria or mesenteric lymph nodes and cDC2 frequencies rather declined. DC-specific RelB deficiency led also to a defect of DC development in cryptopatches and around isolated lymphoid follicles in the lamina propria. A shift in the cDC1 to cDC2 ratio in resident DCs of mesenteric lymph nodes was observed.

This is in contrast to what has been reported for peripheral lymph nodes draining the skin, where RelB deficiency in DCs led to an increased frequency of some migratory DC subsets. This increase of tolerogenic migratory DCs was presumed to induce elevated levels of tTreg in the skin-draining lymph nodes. Here, the data indicate that the DC-specific RelB defect led to a selective reduction of cDC2 in the mesenteric lymph nodes but not lamina propria although elevated Treg frequencies were found in both organs. This indicates that the shift to cDC2 is not responsible for the elevated Treg frequencies but correlated with the loss of cluster 3 DC as revealed from scRNA-seq. Whether and how cluster 3DCs contribute to increased Treg frequencies remains open.

The statement from line 190 that "LT β -driven activation of the alternative NF- κ B pathway including activation of RelB is mandatory for the differentiation of a small CP and ILF surrounding cDC subset in the lamina propria of the small intestine" appears overstated without direct functional evidences such as by interfering with LT-beta activation.

Previously, it had been established that RelB expressing migratory CD11c+ CCR7+ DC in healthy mice represent tolerogenic DC transporting and presenting self-antigens in draining lymph nodes where they either expand Foxp3+ tTreg (Ref 26) or induce Foxp3+ pTreg (Ref 25). Such tolerogenic migratory DC have also been shown to express intermediate levels of costimulatory and they were critical for the protection from autoimmunity in the Th1/Th17 EAE model (Ref 27). Here, the authors extend on these findings by showing that such tolerogenic DC exert protection under Th2 conditions as shown by helminth infection and allergy.

Their scRNA-seq data confirm several genes/molecules (mainly CCL22, CCL17/TARC, CD200, Itgb8), that have been identified for Treg-inducing tolerogenic migratory DC before. However, which of these molecules would be responsible for the tolerogenic effect in the intestine or the used disease conditions, remains open. The RelB-dependent signature (Fig. 2f) also indicates that a loss of RelB increases tolerogenic transcripts such as Itga8 and Ccl17 but also down-regulated others such as CD200, CD63 and Aire. These interesting findings could indicate a specific role of some increased transcripts/markers. Since they were not further functionally investigated, these data remain descriptive by reporting published markers and do not contribute to explain the higher Treg frequencies.

It has also been shown before that RelB is specifically essential for the CD117+ CD172a+ cDC2 subset development in mice and subsequent type 2 cytokine production of IL-4 and IL-13 (<https://doi.org/10.1002/eji.201747332>). However, this paper is not cited. Taken this into consideration, the finding that lower Th2 responses in the Th2 infection model may be also due to lower frequencies or activity of Th2-inducing DC2, rather than or in addition to higher Treg frequencies.

For the scRNA-seq analysis, CD45+ CD64- B220- CD11c-hi MHC-II+ cells were pre-sorted to identify DCs. This biased cell sorting before sequencing excludes potentially interesting CD64+ monocyte derived DC that have been shown before to contribute to Th2 immunity in a house dust mite allergy model (<http://dx.doi.org/10.1016/j.immuni.2012.10.016>).

It remains unexplained why or how the allergic mice are protected from anaphylactic shock despite elevated Th2 cells, IgE levels but no further changes in DCs, mast cells and Treg frequencies. Activation states of tolerogenic DCs or Treg were not tested which could have helped to substantiate the protective activity of these cell types under the challenging conditions. Reasons or mechanisms for the protection are not provided.

Overall, the large amount of accurately obtained and nicely displayed data, adds little new biological information on tolerogenic DCs and the role of RelB for DCs. The increased frequencies of Treg by DC-specific RelB expression remains unexplained. The reasons for protection of the mice under forced Th2 conditions (allergy) or the increased parasitic load (worm infection) correlate with the increased Treg frequencies but without conclusive experiments showing that Treg are responsible for these effects. Treg depletion experiments using anti-CD25 antibody were performed only in worm infected mice. There the experimental groups of untreated and isotype treated RelBDCKO mice were pooled for the figures 5j - 5m and show moderate effects. They should be shown separately to allow a clear distinction between the results of these controls and their significance.

Lines 974 and 977: Change 'C11c+ MHCIIhigh DCs' into CD11c + MHCIIhigh DCs.

Reviewer #3

(Remarks to the Author)

This is an interesting study, linking RelB expression in conventional CD11c+ dendritic cells (cDCs) to their ability to regulate immunity. In particular, the authors present data to indicate that RelB restricts cDC ability to promote/maintain Tregs and Th2 cells, even in helminth infection.

In general, the data presented are of a high quality and are novel and convincing, substantially increasing basic understanding of the role of RelB in regulating DC ability to coordinate immune responses in vivo. As such, the results presented in this study will be of interest to a wide range of researchers.

Major points:

Figure 1 raises the interesting idea that RelB controls cDC ability to promote Foxp3+ Tregs (particularly GATA3+ Tregs) and Foxp3- GATA3+ CD4+ T cells. I think it is important that the authors include cytokine intracellular staining for Th2 cytokines (IL-4 and IL-13) and IL-10 to give better depth of understanding of the true character/effector function of the Foxp3+ GATA3+ and Foxp3-GATA3+ T cells they see increasing in the absence cDC expression of RelB (as they have included in Figure 4). They should also expand their Discussion of this aspect of their data in relation to the literature on Treg stability/flexibility and what the Foxp3+GAT3+ cells they see expanding might represent, in terms of both origin and function (ie 'ex Tregs', or 'ex Th2 cells', or something else?).

It is important for the authors to confirm the diverse DC clusters that they have identified at an mRNA level by scRNAseq at a protein level (ie using multi-parameter flow cytometry or mass cytometry) for Figures 2 and 5.

I'm not sure Figure 3 adds very much to the story, given the marginal impact of cDC RelB deficiency in the presented experiments. I would suggest moving to the Supplement.

It's interesting that DC restricted RelB deficiency doesn't have a dramatic impact on protection against Hpb primary infection, but does impact secondary infection. However, as currently presented, it is difficult to assess what immune changes might have taken place in those primary infection experiments that differ from the secondary infections. The authors should include data on Th2 cells for primary infection to Figure 4, for consistency and to enable comparison to secondary infection data (ie the same readouts as presented in Figure 4 g and i for primary infection experiments). Fig 4J could be moved to the Supplement to provide space for this. Possible explanations for this lack of impact of DC RelB deficiency on primary Hpb infection should be more fully addressed in the manuscript Discussion.

Minor points:

Some of the main Figures are over-busy and cluttered and would benefit from judicious reduction/reformatting and movement of data not vital for the main takehome message into a supplement.

The grammar needs attention throughout the manuscript.

Version 1:

Reviewer comments:

Reviewer #1

(Remarks to the Author)

The authors answered the most parts of concerns and additional experiments improved the overall quality of the manuscript. In addition, the reviewer accepted some technical difficulties. Despite this, I have still major concern that dampen my enthusiasm for this paper which I feel must be addressed.

In the revision, how deficiency of RelB in CD11c+DCs leads to the enhanced proportion of Foxp3+ Treg cells, probably derived from thymic Treg, remains unclear.

Because the data of scRNAseq gene expression are descriptive, the mechanistic insight should be mentioned. While the authors insisted that "it is not possible to pinpoint the effect of RelB resulting in an increase of Tregs on one specific mechanism. Multiple effects may need to come together to induce such a sustained increase in Tregs..., they should address the mechanisms and/or discuss them more precisely in detail.

Reviewer #2

(Remarks to the Author)

The revisions in the manuscript and the responses of the authors have sufficiently clarified my points.

Reviewer #3

(Remarks to the Author)

This is an improved version of the original manuscript that adequately addresses most of my comments.

However, the main Figures remain over-busy, with way too many panels per Figure. The sheer quantity of data per Figure makes reading the manuscript a chore, and reader identification of the core result/main message from each Figure extremely challenging. Scientific publications should be written in as clear, engaging and accessible a manner as possible, with data judiciously selected and presented to achieve this basic goal (which the current manuscript fails to achieve). The data onslaught presented by the authors fails to clearly convey what the main/interesting points and take-home message for each Figure are. As per my original feedback, I would strongly advise judicious reduction/reformatting of all Figures to improve the clarity of the core results throughout this manuscript, with movement of data not vital for the main message in each section moved to supplementary Figures, wherever possible. Without this, the manuscript will struggle to engage anyone other than an extremely specialist reader.

Version 2:

Reviewer comments:

Reviewer #1

(Remarks to the Author)

The authors answered my concern.

Reviewer #3

(Remarks to the Author)

The authors have done a very good job of judiciously reducing/reformatting all Figures, which I think has really improved the clarity of the core results of the manuscript.

REVIEWER COMMENTS

We thank the reviewers for their time and thorough review of our manuscript that substantially helped to improve and strengthen the clarity of our manuscript. A detailed color-coded pt-to-pt reply is provided below.

Reviewer #1 (Remarks to the Author):

The manuscript by Geiselhöringer et al examines the role of RelB in the development and functions of several intestinal cDC subsets by using *ItgaxCre x RelB^{fl/fl}* (RelB Δ DC) mice. RelB Δ DC mice showed an increase in total Foxp3⁺ Treg cells, including GATA3⁺Foxp3⁺ Treg cells, in mesenteric lymph nodes (mLN) and the lamina propria of the small intestine (SI-LP). The development of CIA-DCs seems to depend on RelB as CIA-DCs were completely absent in SI-LP of RelB Δ DC mice. Although RelB Δ DC mice harbored almost normal frequency and cell numbers of cDC subsets in mLN and SI-LP, migratory cDCs in mLN of RelB Δ DC mice exhibited the gene signatures favoring DC-Treg cell interaction by affecting the expression of chemokines, migration behavior, co-stimulatory molecule, and tolerance-related integrin expression. RelB Δ DC mice exhibited the mild anaphylactic reaction despite of the elevated allergic immune responses in a model of food allergy. While RelB Δ DC mice failed to mount a protective Th2 immune response against secondary Hpb infection, the depletion of Foxp3⁺ Treg cells by anti-CD25 mAb restored this response in RelB Δ DC mice. Of the results in this manuscript, the finding that RelB controls intestinal DCs is perhaps the novel and interesting, however mechanistic evaluation or a direct link between this and Foxp3⁺ Treg cells in type 2 immunity is lacking. Therefore, the manuscript, in its current state, raises several questions.

General:

The authors showed the impact of the deficiency of RelB on the development and functions of several intestinal cDC subsets. Can the deficiency of RelB affect the development and functions of cDC subsets in other tissues? How RelB regulates the development of CIA-DCs from cDC progenitors? RelB Δ DC mice displayed the enhanced proportion of Foxp3⁺ Treg cells, while they lacked CIA-DCs. It is not clear to this reviewer how CIA-DCs controls Foxp3⁺ Treg cells. The authors should comprehensively characterize this intestinal cDC subset, including immunogenic to generate Th cells and suppress Treg cells or tolerogenic to generate Treg cells, in a model of type2 immunity. Although the authors analyzed the gene expression profiles of migratory cDCs in mLN in RelB Δ DC mice by scRNAseq, it remains unclear how the deficiency of RelB regulates their functions leading to the enhanced proportion of Foxp3⁺ Treg cells to suppress type 2 immunity, including anaphylactic reaction in food allergy and Hpb infection in mice. In the Results, several speculations were found in each paragraph without any direct evidences, including Line 188-192, Line 233-235, Line 264-267, and Line 363-366. In addition, several misprints/abbreviations/syntax problems were detected, including lines 51, 60, 146, 146-147, and 166, among others.

The reviewer raises several important questions here that we want to answer in the following:

1. Can the deficiency of RelB affect the development and functions of cDC subsets in other tissues?

Yes, this is definitely the case and this was shown for skin (ref 26) and spleen (ref 30). Nevertheless, a detailed functional analysis has not been done yet. To gain deeper insight into system changes in DCs upon RelB depletion, we have initially performed a bulkRNAseq screen of DC subsets isolated from spleen, lung, SI-LP, colon and mLN (cDC1, cDC2, for mLN were even further subdivided into resident and migratory DCs based on gatings shown in Figure 1/Supplementary Fig. 7). As shown below, we observed the highest numbers of differentially expressed genes (DEGs) between RelB Δ DC and control mice in cDC subsets isolated from the mLN (see Fig. 1 in pt-to-pt reply below). Combined with the fact that regulatory T cells in the intestine play a special role in maintaining the tolerogenic

environment, we decided to focus on the intestinal tract and draining mesenteric lymph nodes. However, as our scRNAseq data cannot be compared to such bulk data due to the different definitions of cDC subsets for sorting and cluster annotations, we decided not to include bulk RNAseq data. Nevertheless, these data support the key role of RelB as a transcriptional regulator of cDCs in the mLN and motivated us to focus our analysis on the gut and draining mesenteric lymph nodes.

Fig. 1: Bulk-RNAseq analysis of individual cDC subsets isolated from various organs. Shown are number of DEGs of respective cDC subsets from RelB^{ADC} relative to WT cells.

- How RelB regulates the development of CIA-DCs from cDC progenitors? RelB^{ADC} mice displayed the enhanced proportion of Foxp3⁺ Treg cells, while they lacked CIA-DCs. It is not clear to this reviewer how CIA-DCs controls Foxp3⁺ Treg cells. The authors should comprehensively characterize this intestinal cDC subset, including immunogenic to generate Th cells and suppress Treg cells or tolerogenic to generate Treg cells, in a model of type2 immunity.

The question of how CIA-DCs differentiate from DC progenitors was not the main focus of this study but will be an important question for the future. Moreover, we do not claim in our manuscript that CIA-DCs are responsible for the enhanced proportion of Tregs. To better understand the connection between CIA-DCs and Tregs, additional experiments were performed using CD11c^{Cre} x LtbR^{flox/flox} mice (shown in new Fig. 2d,e) indicating that the sole absence of CIA-DCs in these animals does not result in a similar increase of Tregs or the relative distribution of GATA3- and ROR γ t-expressing Tregs as found in RelB^{ADC} mice (shown in new Fig.2f-i). Moreover, we confirmed a strong reduction/absence of CIA-DCs by flow cytometry according to a recent gating strategy for their bona-fide identification (PMID: 33207209) shown in new Fig.2c.

- Although the authors analyzed the gene expression profiles of migratory cDCs in mLN in RelB^{ADC} mice by scRNAseq, it remains unclear how the deficiency of RelB regulates their functions leading to the enhanced proportion of Foxp3⁺ Treg cells to suppress type 2 immunity, including anaphylactic reaction in food allergy and Hpb infection in mice.

We have validated some of the possible pathways (e.g. CCL22) on protein level but as RelB acts as a transcription factor and as such affects the regulation of multiple genes/pathways, it is not possible to pinpoint the effect of RelB resulting in an increase of Tregs on one specific mechanism. Multiple effects may need to come together to induce such a sustained increase in Tregs that to our knowledge has not been observed in any other models investigating the role of DCs for Treg homeostasis. This highlights the power of RelB as a transcription factor to alter this 'Treg set point' to a level still allowing efficient immunity, e.g. after Hpb infection.

Major comments:

- Fig.1: It remains unclear why GATA3-expressing Treg and T helper cells increased in mLNs and SI-LP of RelB^{ADC} mice. The authors also should provide flow cytometry plots,

frequencies, and, total numbers in each panel to appreciate the changes in the composition of immune cells between RelB Δ DC mice and control mice.

Flow cytometry plots and total numbers are now provided in Supplementary Fig. 1. Please note that absolute numbers of cells isolated from the intestinal lamina propria can vary due to isolation efficiency, have therefore often a high variability and should be thus interpreted with care.

2) Fig.2: CIA-DCs were completely absent in SI-LP of RelB Δ DC mice. How RelB controls the development of CIA-DCs? The authors indicate the connection of LT β signaling and CIA-DCs. Thus, the authors should address this by examining similar responses in Itgax^{Cre} x Lt β receptor fl/fl and/or Lt β receptor KO mice. The authors indicated that the migratory cDC subsets in mLNs of RelB Δ DC mice exhibited the gene signatures favoring DC-Treg cell interaction. However, they did not show the functional differences between control cDCs and RelB-deficient cDCs. The authors should directly address the capacity of each intestinal cDCs in both control mice and RelB Δ DC mice to generate Foxp3⁺ Treg cells from Foxp3⁻ T cells in the presence or absence of active and latent form of TGF- β in vitro. This would provide more informative data than gene expression profiles alone. Total numbers should be provided in panels of Fig.2h and Fig.2i.

As outlined above, it was not the focus of this study to assess how RelB regulates CIA-DCs. Furthermore, we provide now direct evidence that neither Treg numbers nor Treg subset distribution are affected in Itgax^{Cre} x LT β R^{flox/flox} mice (shown in new Fig.2d-i and in the text lines....). Thus, the absence of CIA-DCs in Itgax^{Cre} x LT β R^{flox/flox} does not result in a similar Treg phenotype as in Itgax^{Cre} x RelB^{flox/flox} mice and is therefore not causative for the increase in Tregs in Itgax^{Cre} x RelB^{flox/flox} mice.

Which of the different DC clusters regulate Tregs is currently not possible to address for several reasons: First, migratory DC clusters identified by scRNAseq can currently not be isolated by flow cytometry and do not correlate to 'classical' cDC subsets defined by flow cytometry. Second, any in vitro culture experiments will not take into account the high complexity of how RelB in DCs may regulate their interaction with Tregs, e.g. DC migration, spatial organization, immunological synapse formation or duration of DC-Treg interaction times (e.g. via enhanced release of CCL22). Third, we want to clarify that we don't claim that the increased Tregs are derived from de novo-induced Tregs. Rather, based on their transcriptional profile (Fig. 1 and PMID: 31578269) accumulated Tregs are derived from thymic Tregs that possess increased proliferation capacity (PMID: 31578269) and acquire a tissue Treg profile in Itgax^{Cre} x RelB^{flox/flox} mice. Tissue Tregs differentiate from so-called tissue-Treg progenitors in secondary lymphoid organs via a defined differentiation process described by Delacher et al. (PMID: 31924477). Accordingly, we measured Treg precursors for non-lymphoid tissue Tregs and indeed found increased frequencies of these precursors in mLN (shown in new Fig.1g-i). We hope this reviewer agrees that based on these data enhanced de novo Treg induction in Itgax^{Cre} x RelB^{flox/flox} mice is a very unlikely scenario to explain the increase in Tregs, therefore in vitro induction assay would not be informative. The text has been changed accordingly to make this clearer in results and discussion.

Finally, we don't believe that the total numbers for prevFig. 2h and Fig. 2i are helpful because these plots indicate the upregulation of potential regulatory molecules on cDC subsets but do not define individual DC subsets.

3) Fig.3: It is unclear to this reviewer why RelB Δ DC mice did not exhibit similar anaphylactic reaction to control mice, while they also manifested the elevated allergic immune responses in a model of food allergy. Can the elevated proportion of Foxp3⁺ Treg cells contribute to inhibit the anaphylactic reaction in RelB Δ DC mice? If so, how Treg cells suppress the pathogenesis of anaphylactic reaction without inhibiting allergic immune responses? In addition to frequencies, the authors also should provide flow cytometry plots and total numbers in panels of Fig.3d-j.

We want to make clear that $\text{Itgax}^{\text{Cre}} \times \text{RelB}^{\text{flox/flox}}$ mice still developed an anaphylactic reaction even though with a tendency of lower intensity. It is not entirely clear to what this reviewer refers to with the term 'elevated allergic immune response'. We assume the reviewer refers to the increased levels of allergen-specific IgE (previous Fig. 3c). Indeed, non-treated $\text{Itgax}^{\text{Cre}} \times \text{RelB}^{\text{flox/flox}}$ mice already have slightly elevated levels of serum IgE (PMID: 31578269) and the increase in allergen-specific IgE after food allergy may just be a reflection of this bias. However, the Eisenbarth lab recently provided evidence that specific Tfh cells are required for the induction of high-affinity allergen-specific IgE (and not Th2 cells) (PMID: 31371561 and PMID: 32385053). It is thus possible that also Tfh cells in this model are regulated by excessive Treg numbers, e.g. by so-called follicular regulatory T (Tfr) cells but this was not the focus of this study.

Tregs, in general, have multiple ways to directly and indirectly regulate both the sensitization and also innate effector cells responsible for the anaphylactic shock reaction: We performed and included the food allergy experiment because the increase in Tregs phenotypically resembled pathogenic Gata3^+ Tregs that were shown to be reprogrammed into IL-4/IL-13 producing Th2-like cells resulting in mast cell expansion and their enhanced activation during food allergy (PMID: 25769611). Furthermore, Tregs were also shown to inhibit ILC2s in vitro and in vivo (PMID: 27177780). Lastly, Tregs can directly inhibit the activation/proliferation of naïve and/or pro-allergic T and B cells (e.g. PMID: 27596705 and PMID: 31477921). Based on the relatively mild effect we found in food allergy, we decided to prove the functional role of Tregs rather in the Hpb infection model (previous Fig. 4 and 5). Nevertheless, we wanted to exclude a pathogenic role of Tregs phenotypically resembling pathogenic Il4raF709 Tregs in the context of food allergy (PMID: 25769611). This information was included in the current manuscript because it may be valuable information to the scientific community that exploiting the RelB pathway for therapeutic purposes does not result in an enhanced risk for allergy as may be expected from the Il4raF709 model.

Flow cytometry plots and total numbers for prev. Fig. 3d-j are now provided in Supplementary Fig.4 i-k.

4) Fig.4: $\text{RelB}\Delta\text{DC}$ mice failed to mount a protective Th2-driven immune response upon secondary Hpb infection. Furthermore, $\text{RelB}\Delta\text{DC}$ mice showed the elevated number of Treg cells during infection and their Treg cells did not acquire Th2-like capacities despite elevated GATA3 expression and a strong Th2-biased microenvironment. However, the causation is not established with the presented study. In panels of Fig.4h-j, the authors should provide flow cytometry plots and total numbers.

The causation is addressed in prev. Fig.5. Flow cytometry plots and total numbers for Fig. 3d-j are now provided in Supplementary Fig. 5d-k.

5) Fig.5: The authors used anti-CD25 mAb to deplete Foxp3^+ Treg cells in $\text{RelB}\Delta\text{DC}$ mice. However, the use of this mAb seems to be problematic since it could deplete Foxp3^+ Treg cells as well as effector T cells during the immune responses as described elsewhere in the published reports. To precisely analyze the role of Foxp3^+ Treg cells, it could be better to use $\text{RelB}\Delta\text{DC}$ mice \times DREG (depletion of regulatory T cells) mice (J Exp Med. 2007;204:57-63). Although treatment of $\text{RelB}\Delta\text{DC}$ mice with anti-CD25 mAb caused an insufficient depletion of Foxp3^+ Treg cells, but it restored a protective Th2 response after secondary Hpb infection. They indicated "Thus, RelB deficiency in CD11c^+ cells causes a systemic increase in Foxp3^+ Treg cells that result in a dominant immune tolerance and limits immunity to Hpb infection ..." This claim is not supported by the provided data and there is also a lack of cited evidence to back this up. For example, the suppressive activity of Foxp3^+ Treg cells from $\text{RelB}\Delta\text{DC}$ mice against the activation of Th2 cells from Hpb-infected control mice in vitro. Alternatively, the suppressive effect of adoptive transfer with Foxp3^+ Treg cells from $\text{RelB}\Delta\text{DC}$ mice against the protective Th2 immune response against secondary Hpb infection in control mice. To prove the crucial role of dominant immune tolerance mediated via bystander Foxp3^+ Treg cell activity for preventing a protective Th2 immune response against Hpb infection in $\text{RelB}\Delta\text{DC}$ mice, the authors should consider performing such

experiments. In addition to frequencies, the authors should provide flow cytometry plots and total numbers in panels of Fig.5m.

We agree with the reviewer that depletion of regulatory Tregs by using DERE mice would be much more efficient even though also in this model, some Tregs will escape the depletion with time (PMID: 21287334). An intercross of $Itgax^{Cre}$ x $RelB^{lox/lox}$ would be extremely time-consuming given that first individual parent lines need to be backcrossed as these lines need to be kept separately to avoid germline deletion frequently occurring when using the $Itgax^{Cre}$ line. Additionally, the complete absence of Tregs during Hpb has been described to result in an immunological chaos and prevents efficient helminth clearance while anti-CD25-mediated partial Treg reduction in early phases of infection resulted in increased T helper cell responses and worm clearance (PMID: 26286232). In our view, it is therefore even more informative to only reduce Treg frequencies to wildtype levels instead of a complete depletion. We were first rather worried that anti-CD25 would also affect Th2 effector cells and therefore prevent anti-parasite immunity but as this was apparently not a major issue we are confident that the transient normalization of Treg frequencies to a wildtype level is sufficient to restore effective immunity after secondary Hpb infection. We have modified the indicated citation and included more references to back this up.

We are currently not able to isolate Th2 effector cells from Hpb-infected animals due to the lack of a respective reporter. Moreover, we believe that the suppressive capacity of accumulated Tregs in $Itgax^{Cre}$ x $RelB^{lox/lox}$ mice does not specifically control only Th2 effector cells (and we never claim this in the manuscript) but is operative for all type of T cell responses. Accordingly, we have previously shown in the autoimmune EAE model that accumulated Tregs in $Itgax^{Cre}$ x $RelB^{lox/lox}$ animals can suppress (in this case pathological) Th1/Th17 responses as well (PMID: 31578269). Thus, Tregs may readily suppress the initiation of T helper cell responses in general (which in our case results in a Th2-dominated T helper cell response). Additionally, we want to emphasize that we have already shown in a previous publication that the suppressive capacity of Tregs isolated from $Itgax^{Cre}$ x $RelB^{lox/lox}$ animals does not differ from the suppressive capacity of control animals (ref 27). Thus, accumulated Tregs may in general suppress T cell proliferation irrespective of T cell differentiation enforcing the notion that RelB in DCs acts as a rheostat to regulate (tissue) Treg numbers and thus general T cell immune reactivity.

Flow cytometry plots and total numbers for Fig. 3d-j are now provided in Supplementary Fig. 6e-g.

Minor comments:

- 1) Line 120-141: The description in the paragraph is too long and redundant with Results. The paragraph has been shortened to provide a concise summary of the research findings.
- 2) Line 51: Abbreviate type 2 T helper cells (Th2) cells. Please check Th in the manuscript throughout.
Corrected throughout manuscript.
- 3) Line 60: Correct T helper cell to Th cell.
Corrected.
- 4) Line 146: Correct T helper cells to Th cells.
Corrected.
- 5) Line 146-147: Abbreviate mesenteric lymph nodes (mLN) and the lamina propria of the small intestine (SI-LP). In Figures, authors indicated mLN and SI-LP. They are abbreviated in Line 944-945. Please check the manuscript throughout.
Corrected throughout manuscript.
- 6) Line 166: migratory DCs are CD11cintMHC-IIhi. See Line 959.
Corrected.
- 7) Line 238: Data for GATA3+ Th2 cells is indicated in Fig.1e.
Corrected.
- 8) Line 415: DC subset surrounding cryptopatches and isolated lymphoid follicles (CIA-DCs) is abbreviated before (L183).

Corrected.

9) Line 432-433: RelB-deficient DCs expressed lower levels of co-stimulatory molecules (Cd80, Cd86, Cd200)... Fig.2f shows RelB-deficient DCs expressed higher or lower levels of CD80 and CD86 or CD200.

Corrected

10) Line 443: Indicate the words for the abbreviation of mregDCs.

Included.

11) Line 944-945: mesenteric lymph nodes (mLN) and the lamina propria of the small intestine (SI-LP).

Corrected.

12) Line 952: correct T helper cells to Th cells. In manuscript, the authors used Th2 cells and Th17 cells elsewhere.

Corrected.

13) Line 957: DPDC2s (CD103+CD11b+) and SPDC2s (CD103-CD11b+). Indicate the words for the abbreviation of DP (double positive) and SP (single positive).

Corrected.

14) Line 974, 977, Correct C11c+MHCIIhigh DCs and CD11c+MHCIIhi DCs to C11c+MHCIIhigh DCs and CD11c+MHCIIhi DCs.

Corrected.

15) Line 983-985, Correct e, f to h, i.

Corrected.

16) Line 993: ovalbumin (OVA) is abbreviated before (Line 244).

Corrected.

17) Line 1001: expression in Foxp3+ CD4+ Tregs (i) in mLN. Fig.3h, I indicate the date for SI-LP. Correct mLN to SI-LP.

Corrected.

18) Line 1020: correct T helper cells to Th cells.

Corrected.

19) Line 1023: k Quantification of percentage of resident... Fig.4j indicates the data. Correct K to j.

Corrected.

20) Line 1031: Correct comparison (f, h, k) to comparison (f, h, j).

Corrected.

21) Check the Figure legends of supplementary figures as described above.

Corrected accordingly.

Syntaxes/misprints/abbreviations have been corrected. Some of the mentioned 'speculations' in the results sections are meant to provide some background information, summaries of the data or serve to put the data in context and are therefore kept for the readability of readers who are not experts in this field.

Reviewer #2 (Remarks to the Author):

The authors investigated the effect of specific RelB deficiency in CD11c+ DCs on regulatory T cells in the intestinal tract and their changes after induction of type 2 immunity by helminth or allergy induction.

They find that in mice with DC-specific RelB-deficiency the overall frequency and numbers of both resident and migratory DC are not affected in the lamina propria unlike in mesenteric lymph nodes, where resident cDC2 were decreased. This indicates that cDC2 development is affected RelB-deficiency as has been reported before (<https://doi.org/10.1002/eji.201747332>). No changes were observed for migratory DC subsets in the mesenteric lymph nodes (Fig. 1j). Here, the authors should show whether this is a result of the gating strategy for cDC2 by only including CD11b+ cells. What are then the CD11b- cells in Fig. 1h? Is CD11b simply downregulated on their SPDC2 subset?

The referred literature has been included. CD11b⁻ Cells in Fig. 1h may represent simply a tail of CD11b staining according to the used fluorochrome conjugate. CD11b⁻ cells in mLN are also visible in other publications (e.g. PMID: 37557168). One possibility may be that such CD11b⁻ cells represent DC progenitors but to our knowledge there is currently no direct evidence for this hypothesis.

The data further indicate that RelB deficiency in DCs promoted GATA3⁺ Foxp3⁺ tTreg and GATA3⁺ non-Treg expansion in the lamina propria and mesenteric lymph nodes. Thus, the T cell expansion did not correlate with an increased frequency of resident or migratory DC subsets in the lamina propria or mesenteric lymph nodes and cDC2 frequencies rather declined. DC-specific RelB deficiency led also to a defect of DC development in cryptopatches and around isolated lymphoid follicles in the lamina propria. A shift in the cDC1 to cDC2 ratio in resident DCs of mesenteric lymph nodes was observed.

This is in contrast to what has been reported for peripheral lymph nodes draining the skin, where RelB deficiency in DCs led to an increased frequency of some migratory DC subsets. This increase of tolerogenic migratory DCs was presumed to induce elevated levels of tTreg in the skin-draining lymph nodes. Here, the data indicate that the DC-specific RelB defect led to a selective reduction of cDC2 in the mesenteric lymph nodes but not lamina propria although elevated Treg frequencies were found in both organs. This indicates that the shift to cDC2 is not responsible for the elevated Treg frequencies but correlated with the loss of cluster 3 DC as revealed from scRNA-seq. Whether and how cluster 3DCs contribute to increased Treg frequencies remains open.

We want to point this reviewer to the fact that the reduction of cDC2 was only observed for resident cDCs in mLN. Skin- and gut-draining LNs contain very different DC subsets and the Treg inducing capacities of DCs from both sites may differ quite considerably and it is thus not possible to directly compare the two sites.

We now added data of Itgax^{Cre} x LTβR^{flox/flox} mice that selectively lack CIA-DCs (cluster 3 in SI-LP, described in PMID: 33207209) which do not show an increase in Tregs and/or Treg subsets (shown in new Fig. 2d-i). These data clearly indicate that the sole absence of CIA-DCs is not causative for the observed effects on Tregs (see also response to reviewer 1).

The statement from line 190 that "LTβ-driven activation of the alternative NF-κB pathway including activation of RelB is mandatory for the differentiation of a small CP and ILF surrounding cDC subset in the lamina propria of the small intestine" appears overstated without direct functional evidences such as by interfering with LT-beta activation.

We agree that this sentence is an overstatement and have removed it.

Previously, it had been established that RelB expressing migratory CD11c⁺ CCR7⁺ DC in healthy mice represent tolerogenic DC transporting and presenting self-antigens in draining lymph nodes where they either expand Foxp3⁺ tTreg (Ref 26) or induce Foxp3⁺ pTreg (Ref 25). Such tolerogenic migratory DC have also been show to express intermediate levels of costimulatory and they were critical for the protection from autoimmunity in the Th1/Th17 EAE model (Ref 27). Here, the authors extend on these findings by showing that such tolerogenic DC exert protection under Th2 conditions as shown by helminth infection and allergy.

Their scRNA-seq data confirm several genes/molecules (mainly CCL22, CCL17/TARC, CD200, Itgb8), that have been identified for Treg-inducing tolerogenic migratory DC before. However, which of these molecules would be responsible for the tolerogenic effect in the intestine or the used disease conditions, remains open. The RelB-dependent signature (Fig. 2f) also indicates that a loss of RelB increases tolerogenic transcripts such as Itga8 and Ccl17 but also down-regulated others such as CD200, CD63 and Aire. These interesting findings could indicate a specific role of some increased transcripts/markers. Since they were not further functionally investigated, these data remain descriptive by reporting published markers and do not contribute to explain the higher Treg frequencies.

We thank the reviewer for this summary. We believe it is a major step forward that RelB can be considered as a master regulator of a whole cascade of genes directly associated to Treg induction/expansion/maintenance. It may not be only one sole factor that can explain the sustained increase of Tregs in *Itgax^{Cre} x RelB^{flox/flox}* mice because multiple effects on gene regulation affecting migration, interaction with T cells, contact duration, strength of TCR stimulation may need to come together. Moreover, our literature research did not reveal any similar (or directly opposite) effect on either one of the RelB signature genes or any other DC-intrinsic effect that would indicate a similar strong effect on Tregs. In our view, this highlights the power of RelB as a transcription factor to affect several pathways in DCs ultimately resulting in the effects on Tregs described here (see also response to reviewer 1). We have changed the text to make this more clear.

It has also been shown before that RelB is specifically essential for the CD117+ CD172a+ cDC2 subset development in mice and subsequent type 2 cytokine production of IL-4 and IL-13 (<https://doi.org/10.1002/eji.201747332>). However, this paper is not cited. Taken this into consideration, the finding that lower Th2 responses in the Th2 infection model may be also due to lower frequencies or activity of Th2-inducing DC2, rather than or in addition to higher Treg frequencies.

The referred literature has now been included. Furthermore, we also analyzed CD117 expression in the mLN and indeed found a similar (slight) reduction in CD117 expression in total cDCs as described in the indicated reference in the spleen (new Fig. 3h-i). However, CD117 expression did not show up in migratory DC subsets and was not differently expressed between *Itgax^{Cre} x RelB^{flox/flox}* and control mice (see Fig. 2 in pt-to-pt reply below). This may be due to the relatively low general expression of the CD117 marker in cDCs on a relatively small fraction of cDCs or due to the absence of a small population of resident cDC2s that was not apparent in scRNAseq data (see also Fig. 1).

We rather observed enhanced expression of Gata3⁺ T cells (Fig. 1) and elevated IgE levels in serum (PMID: 31578269) in *Itgax^{Cre} x RelB^{flox/flox}* mice suggesting that the lower expression of CD117 on cDCs or the absence of a putative (resident) CD117+ cDC subset is not sufficient to prevent such effects in vivo. Moreover, attenuation of Treg frequencies by anti-CD25 were sufficient to re-induce normal Th2 immunity (new Fig. 6).

Fig. 2: UMAP plots of *Ccr7* (left) and *Kit* (right) expression comparing cDCs from control and *Itgax^{Cre} x RelB^{flox/flox}* mice. Please note that the orientation of the axis is inverted when compared to new Fig.3 in the revised manuscript.

For the scRNA-seq analysis, CD45+ CD64- B220- CD11c-hi MHC-II+ cells were pre-sorted to identify DCs. This biased cell sorting before sequencing excludes potentially interesting CD64+ monocyte derived DC that have been shown before to contribute to Th2 immunity in a house dust mite allergy model (<http://dx.doi.org/10.1016/j.immuni.2012.10.016>).

We thank the reviewer for this suggestion. Monocytes or DCs derived of monocytes were not part of the current study but will be considered in future studies. It is also not clear whether a comparable monocyte-derived DC subset may exist in the intestine as this population has only been described in the lung. Additionally, a huge body of literature suggests that cDCs are particularly relevant for Tregs in the intestinal tract and monocytes themselves do not express substantial levels of RelB (according to Immgen database).

It remains unexplained why or how the allergic mice are protected from anaphylactic shock despite elevated Th2 cells, IgE levels but no further changes in DCs, mast cells and Treg frequencies. Activation states of tolerogenic DCs or Treg were not tested which could have helped to substantiate the protective activity of these cell types under the challenging conditions. Reasons or mechanisms for the protection are not provided.

See related response to Reviewer 1. We want to stress that $\text{Itgax}^{\text{Cre}} \times \text{RelB}^{\text{flox/flox}}$ mice are not protected from food allergy/anaphylactic shock but just a tendency to have less severe symptoms. With this figure, we just wanted to rule out the risk of whether therapeutically targeting RelB in DCs may result in an enhanced risk for allergic reactions which was not the case. We then decided to focus on protection from a secondary Hpb infection as an in vivo model because it is well known that protection relies on the generation of Th2 cells in this model. Tregs in general do have multiple ways to control allergic responses as indicated in the response to a related question of Reviewer 1.

Overall, the large amount of accurately obtained and nicely displayed data, adds little new biological information on tolerogenic DCs and the role of RelB for DCs. The increased frequencies of Treg by DC-specific RelB expression remains unexplained. The reasons for protection of the mice under forced Th2 conditions (allergy) or the increased parasitic load (worm infection) correlate with the increased Treg frequencies but without conclusive experiments showing that Treg are responsible for these effects. Treg depletion experiments using anti-CD25 antibody were performed only in worm infected mice. There the experimental groups of untreated and isotype treated RelBDCKO mice were pooled for the figures 5j - 5m and show moderate effects. They should be shown separately to allow a clear distinction between the results of these controls and their significance.

As outlined in the response to reviewer 1, it is unlikely that a sole molecule is responsible for the observed effects on Tregs and the power of RelB as a transcription factor to regulate multiple pathways that result in such a sustained increase in Tregs maintained even under highly inflammatory conditions is in our view unique. In fact, $\text{Itgax}^{\text{Cre}} \times \text{RelB}^{\text{flox/flox}}$ mice can be genotyped by their Treg frequency, and we are not aware of any other mouse line targeting DCs to have a similar strong effect on Tregs and immune tolerance. We performed the anti-CD25 experiments only in worm-infected $\text{Itgax}^{\text{Cre}} \times \text{RelB}^{\text{flox/flox}}$ mice because there is no clear effect that can be expected to be different post attenuation of Treg levels in the food allergy model.

We believe it is remarkable that the transient attenuation of Tregs in $\text{Itgax}^{\text{Cre}} \times \text{RelB}^{\text{flox/flox}}$ mice to wild-type levels during primary infection is sufficient to restore protective immunity to a secondary infection. In our view, this is a strong in vivo argument that Tregs are responsible for the altered memory response in $\text{Itgax}^{\text{Cre}} \times \text{RelB}^{\text{flox/flox}}$ mice.

The experimental groups of untreated and isotype treated $\text{Itgax}^{\text{Cre}} \times \text{RelB}^{\text{flox/flox}}$ mice in new Fig. 6 are now displayed separately.

Lines 974 and 977: Change 'C11c+ MHCIIhigh DCs' into CD11c + MHCIIhigh DCs.
Corrected.

Reviewer #3 (Remarks to the Author):

This is an interesting study, linking RelB expression in conventional CD11c+ dendritic cells (cDCs) to their ability to regulate immunity. In particular, the authors present data to indicate that RelB restricts cDC ability to promote/maintain Tregs and Th2 cells, even in helminth infection.

In general, the data presented are of a high quality and are novel and convincing, substantially increasing basic understanding of the role of RelB in regulating DC ability to coordinate immune responses in vivo. As such, the results presented in this study will be of

interest to a wide range of researchers.
We thank the reviewer for this very positive statement.

Major points:

Figure 1 raises the interesting idea that RelB controls cDC ability to promote Foxp3+ Tregs (particularly GATA3+ Tregs) and Foxp3- GATA3+ CD4+ T cells. I think it is important that the authors include cytokine intracellular staining for Th2 cytokines (IL-4 and IL-13) and IL-10 to give better depth of understanding of the true character/effector function of the Foxp3+ GATA3+ and Foxp3-GATA3+ T cells they see increasing in the absence cDC expression of RelB (as they have included in Figure 4). They should also expand their Discussion of this aspect of their data in relation to the literature on Treg stability/flexibility and what the Foxp3+GAT3+ cells they see expanding might represent, in terms of both origin and function (ie 'ex Tregs', or 'ex Th2 cells', or something else?).

We have now added intracellular cytokine staining of T helper cells and Tregs at steady-state in the new Supplementary Fig 1i,j. These data do not indicate enhanced type 2 cytokines at steady state despite elevated Gata3-expressing T helper cells. Given that even after Hpb infection we did not observe a similar increase in type 2 cytokine positive T cells compared to the frequency of GATA3+ T cells, T cells of Itgax^{Cre} x RelB^{flox/flox} mice may just have a slight tendency to become Th2 effector cells or a different activation status. As this reviewer suggests it is also possible that Gata3 upregulation is a consequence of transdifferentiated GATA3hi Treg cells that lost Foxp3 but this is at this moment only a speculation. A respective sentence has been added to the discussion as suggested.

It is important for the authors to confirm the diverse DC clusters that they have identified at an mRNA level by scRNAseq at a protein level (ie using multi-parameter flow cytometry or mass cytometry) for Figures 2 and 5.

We have now included a flow cytometry staining to confirm the absence of CIA-DCs according to the definition in a recent paper (PMID: 33207209) shown in new Fig. 2c and in the text lines....). For the mLN clusters, confirmation of clusters is a really challenging issue and, in this case, not really insightful. We want to point out that all clusters were comparable in size between Itgax^{Cre} x RelB^{flox/flox} and control mice. The number of clusters for scRNAseq analysis depends on parameters set during bioinformatic analysis and can thus be reduced or expanded according to settings and needs. As such, there is no direct biological function to each cluster and it may thus not be useful to validate clusters on protein level. On top of that, clustering depends on a variety of differentially expressed genes for each cluster and availability of antibodies limits the capability to identify these clusters beyond a few probably even non-exclusive markers. We have now added on request of reviewer 2 a staining for CD117 now shown in Fig. 3h, i.

I'm not sure Figure 3 adds very much to the story, given the marginal impact of cDC RelB deficiency in the presented experiments. I would suggest moving to the Supplement.

Based on the possible importance of pathogenic (ex-)Tregs co-expressing Gata3 for the development of allergic diseases (PMID: 25769611) we believe this piece of data can be highly interesting to researchers in the allergy field and would like to keep it therefore as a main figure. This data makes clear that the increase of Gata3-expressing Tregs does not go along with an enhanced risk or pathology in food allergy as observed in animals with enhanced IL4R signaling in Tregs and associated enhanced Gata3 expression.

It's interesting that DC restricted RelB deficiency doesn't have a dramatic impact on protection against Hpb primary infection, but does impact secondary infection. However, as currently presented, it is difficult to assess what immune changes might have taken place in those primary infection experiments that differ from the secondary infections. The authors should include data on Th2 cells for primary infection to Figure 4, for consistency and to

enable comparison to secondary infection data (ie the same readouts as presented in Figure 4 g and i for primary infection experiments). Fig 4J could be moved to the Supplement to provide space for this. Possible explanations for this lack of impact of DC RelB deficiency on primary Hpb infection should be more fully addressed in the manuscript Discussion.

We did not measure cytokine secretion because there was almost no difference in the number of Gata3-expressing T cells nor was there a clear difference in parasite burden after primary infection. It is well established that protective immunity in the Hpb model depends on Th2 cells generated during primary infection (PMID: 23053394, PMID: 1683629, PMID, 9355130) while the primary infection results in a chronic infection. We have modified the text accordingly to make this clearer to the reader.

Minor points:

Some of the main Figures are over-busy and cluttered and would benefit from judicious reduction/reformatting and movement of data not vital for the main takehome message into a supplement.

We would be willing to move some of the descriptive data such as frequencies of DC subsets or T cell subsets to the supplement. However, differences in subset distribution at steady state or upon perturbation may have contributed to effects on any of the observed effects so we feel it is still important for the reader to have evidence that this was not the case. Based on this concept and on the request of Reviewer 1 for even additional data on total cell numbers of all subsets (now provided in additional Supplementary Figures), we would like to keep the data in main figures (depending on editorial decision).

The grammar needs attention throughout the manuscript.

Grammar has been checked throughout the manuscript.

REVIEWER COMMENTS

We thank the reviewers for their time and effort to review our manuscript. Please find below a reply to individual comments.

Reviewer #1 (Remarks to the Author):

The authors answered the most parts of concerns and additional experiments improved the overall quality of the manuscript. In addition, the reviewer accepted some technical difficulties. Despite this, I have still major concern that dampen my enthusiasm for this paper which I feel must be addressed.

In the revision, how deficiency of RelB in CD11c+DCs leads to the enhanced proportion of Foxp3+ Treg cells, probably derived from thymic Treg, remains unclear.

Because the data of scRNAseq gene expression are descriptive, the mechanistic insight should be mentioned. While the authors insisted that “it is not possible to pinpoint the effect of RelB resulting in an increase of Tregs on one specific mechanism. Multiple effects may need to come together to induce such a sustained increase in Tregs..., they should address the mechanisms and/or discuss them more precisely in detail.

In the revised manuscript we include new data from additional experiments in which we show that RelB-deficient DCs from mLN have a selective advantage for antigen-specific cluster formation with Tregs at the expense of cluster formation with non-Tregs (new Fig.3f). Importantly, this difference vanishes at later time points demonstrating the principal capacity of RelB-deficient DCs to normally interact with non-Treg cells (new Fig.3g). Given that DC-Treg cluster formation has been shown to strictly depend on Ccl22 (PMID: 30910796) and that RelB-deficient DCs express more Ccl22 (rev. Fig. 3d, e and Fig.1a below), we propose as a mechanism that elevated Ccl22 released by RelB-deficient DCs provides a selective advantage for DC-Treg cluster formation and ‘premium’ access for rare self-peptide:MHC-II complexes. Elevated Ccl22 protein levels can also be found in other lymphoid organs (Fig.1b below) suggesting that similar effects with enhanced DC-Treg cell cluster formation may also be operative in other lymphoid organs. This is in line with the systemic increase in Treg cells in RelB^{ΔDC} animals but as the focus of this manuscript was the intestinal tract, this data has not been included in the manuscript. TCR stimulation of Treg cells is known to be necessary for Treg survival and stimulation of Treg cells via their cognate antigen has been shown to be necessary for the gain of tissue-specific transcriptional signatures (PMID: 29887374). Moreover, Treg cells in the mLN have been recently shown to preferentially interact with migratory DCs (PMID: 36993443; ref now included in the revised manuscript) and Ccl22 is preferentially expressed in migratory DC subsets (see Fig. 1a below and rev. Fig.3d) – the only cDC subsets for which a genotype-specific clustering has been observed in scRNAseq analysis (see Fig. 3a).

Fig. 1 a UMAP plots of scRNAseq analysis depicting the expression of Ccl22 preferentially in migratory cDC subsets (for cluster annotation please refer to Fig. 3a). **b** Quantification of CCL22-levels determined by ELISA in inguinal LN (iLN) and Peyer's Patches (PP) of control and RelB^{ΔDC} mice.

Reviewer #2 (Remarks to the Author):

The revisions in the manuscript and the responses of the authors have sufficiently clarified my points. We thank reviewer 2 for his work to improve our manuscript.

Reviewer #3 (Remarks to the Author):

This is an improved version of the original manuscript that adequately addresses most of my comments.

However, the main Figures remain over-busy, with way too many panels per Figure. The sheer quantity of data per Figure makes reading the manuscript a chore, and reader identification of the core result/main message from each Figure extremely challenging. Scientific publications should be written in as clear, engaging and accessible a manner as possible, with data judiciously selected and presented to achieve this basic goal (which the current manuscript fails to achieve). The data onslaught presented by the authors fails to clearly convey what the main/interesting points and take-home message for each Figure are. As per my original feedback, I would strongly advise judicious reduction/reformatting of all Figures to improve the clarity of the core results throughout this manuscript, with movement of data not vital for the main message in each section moved to supplementary Figures, wherever possible. Without this, the manuscript will struggle to engage anyone other than an extremely specialist reader.

Indeed, it was a difficult decision to find a delicate balance to on the one hand provide enough details and control data but on the other hand don't dilute the main message. We have now removed or moved to the supplement many 'control' data to make the main message clearer. We hope this makes our manuscript better understandable to a more broadly interested readership.